

**High-precision atmospheric oxygen measurement comparisons between a newly built**
**CRDS analyzer and existing measurement techniques**
Tesfaye A. Berhanu[1,2], John Hoffnagle[2], Chris Rella[2], David Kimhak[2], Peter Nyfeler[1], Markus
Leuenberger[1]
[1]*Climate and Environmental Physics, Physics Institute and Oeschger Centre for Climate Change Research,*
*University of Bern, Bern, Switzerland*
[2]*Picarro Inc., 3105 Patrick Henry Drive, Santa Clara, CA, USA*
**Abstract**
Carbon dioxide and oxygen are tightly coupled in land-biospheres $CO_2$ - $O_2$ exchange
processes, while they are not coupled in oceanic exchange. For this reason, atmospheric
oxygen measurements can be used to constrain the global carbon cycle, especially oceanic
uptake. However, accurately quantifying the small (~1-100 ppm) variations in $O_2$ is
analytically challenging due to the very large atmospheric background which constitutes
about 20.9 % (~209500 ppm) of atmospheric air. Here we present comprehensive laboratory
and field studies for a newly developed high-precision oxygen mixing ratio and isotopic
composition analyzer (Picarro G-2207) that is based on cavity ring-down spectroscopy
(CRDS). From the laboratory tests, we have calculated a short-term precision (standard error
of one-minute measurements) of < 1 ppm for this analyzer based on measurements of eight
standard gases analyzed for two hours consecutively. In contrast to the currently existing
techniques, the instrument has an excellent long-term stability and therefore a calibration
every 12 hours is sufficient to get an overall uncertainty of < 5 ppm. Measurements of
ambient air were also conducted at the High-Altitude Research Station, Jungfraujoch and the



Beromünster tall tower in Switzerland. At both sites, we observed opposing and diurnally
varying $CO_2$ and $O_2$ profiles due to different processes such as combustion, photosynthesis
and respiration. Based on the combined measurements at Beromünster tower, we determined
height dependent $O_2$:$CO_2$ oxidation ratios varying between -0.98 to -1.60 , which increase
with the height of the tower inlet, possibly due to different source contribution such as natural
gas combustion with high oxidation ratio and biological processes which are at the lower end.
**1. Introduction**
Atmospheric oxygen comprises about 20.9 % of the global atmosphere and in the past
decade its concentration decreased at a rate of ~ 20 per meg $yr^{-1}$ (Keeling and Manning, 2014)
mainly associated with the increase in fossil fuel combustion. In contrast, the global average
atmospheric $CO_2$ mixing ratio increased to 402.8 ppm averaged over 2016 (predicted to grow
by 2 % in 2017) since its preindustrial value of 280 ppm (Le Quéré et al., 2017). As the
variability of atmospheric oxygen is directly linked to the carbon cycle, both its short and
long-term observations can be used to better constrain the carbon cycle. For example, since
first suggested by Keeling and Shertz (1992) the long-term trends derived from concurrent
measurements of atmospheric $CO_2$ and $O_2$ have been widely used to quantify the partitioning
of atmospheric $CO_2$ between the land-biosphere and oceanic sinks (Battle et al., 2000; Goto et
al., 2017; Manning and Keeling, 2006; Valentino et al., 2008). This method hinges on the
linear coupling between $CO_2$ and $O_2$ with an oxidation ratio (OR) of 1.1 for the terrestrial
biosphere photosynthesis-respiration processes ($\alpha_b$) and 1.4 for fossil fuel combustion ($\alpha_f$)
while they are decoupled for oceanic processes. Meanwhile, the short-term variability in
atmospheric oxygen can be used to estimate marine biological productivity and air-sea gas
exchange (Keeling et al., 1998; Nevison et al., 2012). However, the accuracy of these
estimates is primarily linked to the accuracy and precision of atmospheric $O_2$ measurements





and the assumed ORs for the different processes which are highly variable in contrast to
atmospheric $CO_2$ that can be well measured within the precision guidelines set by the Global
Atmospheric Watch (GAW) ($\pm$ 0.1 ppm for the northern hemisphere).

Currently there are several techniques that can measure atmospheric $O_2$ variations as

oxygen concentration based on interferometric, paramagnetic, UV absorption and fuel cell
technology (Keeling, 1988a; Manning et al., 1999; Stephens et al., 2007) or as $O_2/N_2$ ratios to
account for the large background effect using Gas chromatography with thermal conductivity
detector (GC-TCD) or Gas chromatography coupled to mass spectrometry (GC-MS) (Bender
et al., 1994; Tohjima, 2000). Despite the fact that these techniques have been commercially
available for more than two decades, accurate quantification of atmospheric oxygen
variability remains challenging primarily because the small ppm-level atmospheric oxygen
signal rides on a ~ 210,000 ppm background, which places stringent requirements on the
precision and drift of the analysis methods. The techniques listed above struggle to routinely
achieve the necessary performance for various reasons, including i) instability over time that
requires frequent measurement interruption for calibration, ii) measurement bias with ambient
and sample temperature and/or pressure, and/or iii) systematic errors in the measurement due
to other atmospheric species. Further, some techniques require the use of consumables and
rely on high vacuum, which complicates field deployment.

In this manuscript we describe a new high precision oxygen concentration and isotopic

composition analyzer by Picarro Inc., Santa Clara, USA (G-2207) based on CRDS
technology. Here, we will introduce the analyzer design principles in details, describe the
unique features of the analyzer and evaluate its performance based on various independent
laboratory and field tests by comparing it with currently existing techniques. Then, we will



present and interpret our observations based on field measurements. Finally, we will conclude
its overall performance and provide recommendations and possible improvements.
**2. Analyzer design principles**

The analyzer described here is derived from the Picarro G2000 series of CRDS

analyzers.  The basic elements have been described elsewhere (Crosson, 2008; Martin et al.,
2016; Steig et al., 2014):  briefly, the instrument is built around a high-finesse, traveling-wave
optical cavity, which is coupled to either of two single-frequency DFB-stabilized
semiconductor lasers. One cavity mirror is mounted on a piezoelectric translator (PZT) to
allow fine tuning of the cavity resonance frequencies. A semiconductor optical amplifier
between the laser sources and the cavity boosts the laser power and serves as a fast-optical
switch. The cavity body is constructed of invar and enclosed in a temperature stabilized box
(T = 45° C, stabilized to approximately 0.01 °C) for dimensional and spectroscopic stability.
A vacuum pump pulls the gas to be sampled through the cavity and a proportional valve
between the cavity and the pump maintains the sample pressure in the cavity at a value of 340
hPa, with variations on the order of 1 Pa. The instrument has a wavelength monitor, based
upon measurements of interference fringes from a solid etalon, which is used to control the
laser wavelength by adjusting the laser temperature and current. A high-speed photodiode
monitors the optical power emerging from the cavity. The instrument's data acquisition
system sweeps the laser frequency over the spectral feature to be measured, modulates the
laser output to initiate ring-downs, and fits the ring-down signal to an exponential function to
generate a spectrogram of optical loss versus laser frequency. Subsequent program modules
compare the measured loss spectrum to a spectral model, using non-linear least-squares fitting
(Press et al., 1986) to find the best-fit model parameters and thereby obtain a quantitative
measure of the absorption due to the target molecule, and finally apply a calibration factor to



the optical absorption to deduce the molecular concentration. When operating in its normal
gas analysis mode, the instrument acquires about 200-300 ring-downs per second and
achieves a noise equivalent absorption of typically about $10^{-11}$ cm$^{-1}$ Hz$^{-1/2}$, with some variation
between instruments.
The primary goal when designing this analyzer was to measure the molecular oxygen
concentration with few-per-meg level precision and stability. In this context operational
stability is as important as signal-to-noise. Our experience has been that the most stable
operation of the analyzer is achieved when the optical phase length of the cavity is held as
nearly constant as possible. In this case the free spectral range (FSR) of the temperature
stabilized, invar ring-down cavity provides a better optical frequency standard than the etalon-
based wavelength monitor, which in turn allows more consistent measurements of absorption
line width and integrated absorption line intensity (Steig et al., 2014). For a small, field-
deployable instrument, it is not practical to stabilize the absolute frequencies of the cavity
modes to an optical frequency standard (Hodges et al., 2004) but the oxygen lines themselves,
under conditions of constant temperature and pressure, provide an adequate frequency
reference. The oxygen spectrum was also used to calibrate the FSR, by comparing a wide
(approximately 10 cm$^{-1}$) FSR-spaced spectrum with the Hitran database (Rothman et al.,

2013).

To determine molecular oxygen concentration, the analyzer measures absorption of the
Q13Q13 component of the $a^1\Delta_g \leftarrow X^3\Sigma^-_g$ band, at a frequency of 7878.805547 cm$^{-1}$,
according to the latest edition of Hitran (Gordon et al., 2017). This is one of the strongest
near-infrared lines of oxygen, well separated from other oxygen lines, and reasonably free of
spectral interference from water, carbon dioxide, methane, and other constituents of clean air.
The spectral model for this line was developed using reference spectra of clean, dry, synthetic





air that were acquired with the same hardware as in the field-deployable analyzer, but with
special-purpose software that allows it to operate as a more general spectrometer.

Recently, considerable work has been done to advance the understanding of spectral

line shapes and to define functional representations that better describe the processes that
determine spectral line shapes than does the Voigt model (Hartmann et al., 2008; Tennyson et
al., 2014). Line shape studies have been published for the 1.27 μm band of $O_2$ (Fleisher et al.,
2015; Lamouroux et al., 2014), though not to our knowledge for the Q branch. The apparatus
used here is not capable of spectroscopic studies of comparable precision; the absolute
temperature and pressure monitoring and especially the frequency metrology are far too crude
for that purpose. Our goal is merely to define a simple model of the Q13Q13 line that is
adequate for least-squares retrievals of the $O_2$ absorption under the limited range of conditions
(stabilized temperature and pressure) that the operational analyzer experiences in the field.
The CRDS analyzers use the Galatry function (Varghese and Hanson, 1984), which is
distinctly better than the Voigt and still easily and quickly evaluated for line shape modeling.
Ultimately, the usefulness of the spectral model is to be evaluated by the precision and
stability of the $O_2$ measurements when compared with established techniques. We also note at
this point that Sironneau, Fleisher, and Hodges have made detailed measurements of lines in
the R branch of the $a^1\Delta_g \leftarrow X^3\Sigma^-_g$ band and observed departures from simple, linear
absorption, which they interpret as arising from collision-induced absorption (Fleisher et al.,
2015). This has two important consequences for $O_2$ monitoring: the line strength is not
independent of sample pressure, and optical absorption is not linear in laser intensity. We do
not expect these effects to be too severe for our application because the ring-down cavity is
stabilized to a very narrow range of temperature and pressure. In addition, the optical power
in the ringdown cavity set by the ring-down detector threshold, which is used to trigger the





laser shutoff and subsequent ring-down waveform acquisition. The fact that all ring-downs
occur at the same intracavity power should minimize the effect of collision-induced
absorption. We have observed some excess noise on the ring-down time constants for the
highest loss points at the peak of the Q13Q13 line, which might have to do with the fitting of
the ring-down signal if absorption is not linear, but we cannot be certain of this explanation at
present.

For spectral model development, this spectrometer has the drawback that the cavity

FSR, equal to about 0.0206 cm$^{-1}$, is too large to reveal much detail of the absorption line
shape, even with the simplifying assumption of a Galatry line shape. We therefore acquired a
set of four interleaved spectra, with the PZT-actuated mirror moved to offset the cavity modes
of the individual FSR-spaced spectra by one-fourth of an FSR. The precise offsets were
determined from fits to the strong and well-isolated $O_2$ lines in the spectra. From the
consistency of the fitted line centers, we estimate that the positioning of the interleaved
spectra was accurate to approximately 10 MHz. The spectrum of the Q13Q13 line acquired in
this manner is shown in Figure 1, together with the best-fit Galatry function. It stands out that
the residuals that are largely an odd function of detuning from the line center: this shows the
limitations of the Galatry model in this case, since the Galatry function is purely even about
the line center. The shape of the absorption line in this model is specified by two
dimensionless parameters: the collisional broadening parameter
$y = \gamma / \sigma_D$              (1)
and the collisional narrowing parameter
$z = \beta / \sigma_D$              (2)
where $\gamma$ is the frequency of broadening transitions, $\beta$ is the frequency of narrowing collisions,
and $\sigma_D$ is the Doppler width of the transition, given by





$\sigma_D = \nu_0 (2k_B T/Mc^2)^{1/2}$                                                   (3)
where $\nu_0$ is the transition frequency, $k_B$ is Boltzmann's constant, T is the sample temperature,
M is the molecular mass, and c is the speed of light. Figure 2 shows the values of y and z
obtained from spectra acquired in the same way as Figure 1, as a function of cavity pressure.
The values depend linearly on pressure, as expected from the Galatry model, but the
unconstrained linear fits do not go precisely through the origin. It is not clear whether this
represents a breakdown of the Galatry model or simply reflects the limited quality of the data
set. The slope of y can be converted to an air-broadened collisional width $\gamma_{air} = 0.0442$ cm$^-$
$^1$/atm, which agrees with the Hitran value of 0.0460 cm$^{-1}$/atm (Rothman et al., 2013) to within
the uncertainty estimate stated by Hitran. The slope of z can be interpreted in terms of the
optical diffusion coefficient (Fleisher et al., 2015), yielding D = 0.285 cm$^2$ s$^{-1}$, compared to
the literature value of 0.233 cm$^2$ s$^{-1}$ for $O_2$ in air at 45 °C (Marrero and Mason, 1972).
Although the anticipated use of the analyzer is for ambient air samples having a very small
range of $O_2$ concentrations, we did investigate the variation of the line shape in binary
mixtures of $O_2$ and $N_2$ shown in Figure 3. The error bars are taken from the output of the
Levenberg-Marquardt fitting routine (Press et al., 1992). The dependence of the collisional
broadening parameter z on $O_2$ mole fraction was considered too small to be significant, but
the variation in y was used in the subsequent analysis of the air samples.  Note that Wójtewicz
et al. (Wójtewicz et al., 2014) also found collisional broadening coefficients for nitrogen to be
slightly larger than for oxygen in measurements of one $O_2$ line in the B-band.

The primary goal in designing the analyzer was to achieve high enough precision to

make meaningful measurements of $O_2$ in clean atmospheric samples. Although the current
best practice for such high-precision measurements is to work with dried samples, we decided



to include high precision measurements of water vapor. There were two reasons for this
decision: one is to serve as a monitor for residual water vapor, which is difficult to remove
completely from the ring-down cavity and associated sample handling hardware, and the
second and more ambitious reason was to see how well the effect of water vapor could be
corrected for in measurements of undried ambient air. While it was considered unlikely that
measurements of undried air could compete in accuracy with those of dried air, it might be
possible to correct for water vapor well enough to enable useful measurements in some
circumstances without the expense and inconvenience of drying the sample.  For this purpose,
a second laser was added, which probes the $7_{1,6} \rightarrow 8_{4,5}$ component of the $2v_3$ band of water
vapor, at a frequency of 7816.75210 cm$^{-1}$ (Gordon et al., 2017).  The Galatry model was used
to fit spectra of synthetic air humidified to various levels of water vapor concentration. These
fits also included two other nearby, very weak water lines, with intensities less than 1% of the
strong transition, in order that their absorption should not perturb the line shape of the main
transition.  Results for the shape of the 7816.75210 cm$^{-1}$ line are shown in Figure 4. At the
level that we can measure, only the y-parameter has a meaningful variation with water
concentration. From the linear fit one obtains a pressure broadening coefficient for air, $\gamma_{air}$ =
0.0752 cm$^{-1}$/atm, in reasonable agreement with the Hitran value $\gamma_{air}$ = 0.0787 cm$^{-1}$/atm
(Gordon et al., 2017), and a self-broadening coefficient $\gamma_{self}$ = 0.413 cm$^{-1}$/atm, to be compared
with the Hitran value $\gamma_{self}$ = 0.366 cm$^{-1}$/atm. Since the uncertainty estimate for the Hitran
values is 10 % to 20 %, this level of agreement seems reasonable.

We also looked at absorption from water near the Q13Q13 absorption line of $O_2$.

These spectra were measured in a background of pure nitrogen to reveal the very weak lines
interfering with the $O_2$ measurement. Without the strong $O_2$ lines, it was impossible to
interleave FSR-spaced spectra, so in this case the frequency axis comes from the analyzer's





wavelength monitor. The upper panel of Figure 5 shows the spectrum of saturated water vapor
in nitrogen, together with a fit to a Voigt model of the molecular lines. The measurement was
made at a pressure of 340 hPa and temperature of 45° C. The two most prominent features in
this spectrum are actually the Q17R16 and Q13Q13 lines from traces of $O_2$ remaining in the
sample while the other features are from water. The lower panel of Figure 5 shows the lines
tabulated in Hitran. Immediately after the data in Figure 5 were acquired, measurements were
also made at 7816.85210 cm$^{-1}$, to establish the relationship between the absorption strengths
in the two spectral regions. All the water lines that were observed, in both spectral regions, are
from the dominant 161 isotopolgue of water, so changes in isotopic composition of
atmospheric water does not lead to variation in the relative strengths of the lines we measure.
Hitran simulations for molecules other than water that are expected to be present in clean,
ambient air indicate that direct interference with the Q13Q13 line should be negligible at the
level of precision considered here. In the case of $CO_2$, the dilution of oxygen due to 400 ppm
of $CO_2$ is significant, and larger than any direct spectral interference.

Finally, we investigated the influence of water vapor on the shape of the $O_2$ Q13Q13

line. Switching between the two lasers sources, we acquired FSR-spaced spectra of
humidified synthetic air, alternately covering the 7817 cm$^{-1}$ and 7878 cm$^{-1}$ regions.  Individual
spectra were acquired in less than 2 s, so changes in water vapor concentration between
spectra were small. These spectra, with frequency resolution of 0.0206 cm$^{-1}$, were analyzed by
nonlinear least-squares fitting with the following spectral models:  the 7817 cm$^{-1}$ spectra were
modeled as the sum of an empty-cavity baseline having an adjustable offset level and slope
and three water peaks and the two weak perturbing peaks. The molecular absorption of the
main peak was expressed as an adjustable amplitude, $A_w$, multiplying a dimensionless, area-
normalized Galatry function (Varghese and Hanson, 1984).  The weak perturbers were
modeled by Voigt profiles with amplitudes and line widths that constrained to be in fixed
ratios to the strong line, and therefore added no new degrees of freedom to the fitting
procedure. Since the amplitude $A_w$ multiplies an area-normalized shape function, it is
essentially equivalent to the area of the absorption line, to the extent that the Galatry model
provides a valid description of the line shape. The Doppler width of the Galatry function was
fixed based on the measured cell temperature, the y-parameter was allowed to vary, and the z-
parameter was constrained to be proportional to y, based on the earlier measurements. In
addition, the center frequency of the Galatry function was adjusted to match the data set,
giving a total of five free parameters for this fit. The 7878 cm$^{-1}$ spectra were modeled with an
adjustable baseline offset and slope and molecular absorption amplitude, $A_{O2}$, describing the
Q13Q13 $O_2$ line. Here, too, the y-parameter and centration of the $O_2$ lines were allowed to
adjust, and the z-parameter was constrained to be proportional to y. The weak water lines
interfering with oxygen absorption were included in the model, but with no additional free
parameters, rather the amplitudes were preset based on the measured water absorption at 7817
cm$^{-1}$ and the previously determined amplitude relationships between the water lines.
Collisional broadening of the Q13Q13 $O_2$ line by water vapor is shown in Figure 6. From the
linear fit one obtains a coefficient for collisional broadening of the Q13Q13 line by water
vapor of $\gamma_{water}$ = 0.0442 cm$^{-1}$/atm. We are not aware of previous measurements of this
quantity.
The alternating measurements at 7817 cm$^{-1}$ and 7878 cm$^{-1}$ also calibrated the
relationship between water mole fraction and the absorption at 7817 cm$^{-1}$, using a dilution
analysis described by Filges et al. (Filges et al., 2018), who showed that the results obtained
this way agree well with water vapor fractions measured with a conventional hygrometer.
Figure 7 shows the measured amplitudes of the water and oxygen lines for samples of variable





humidity. Since the air came from a tank of constant composition, the oxygen concentration
changes due to dilution of oxygen when water is added. Assuming that this is the sole cause
of the change in measured absorption, since the line shapes were being constantly adjusted to
account for changes in collisional broadening, it is straightforward to deduce the relation
between the water fraction and the absorption amplitude. This calibration was used to
generate the water fraction axes in Figures 4 and 6. We note that we did not take particular
care to control or measure the quantity of dissolved gases, especially oxygen and carbon
dioxide, in the water used for this experiment. While these gases would not significantly
affect the water calibration, they may affect the water vapor correction of the oxygen
measurement at the ppm level.  More work needs to be done to investigate the water vapor
correction of the oxygen measurement.

The observations described above were used to design a method to measure oxygen

concentration in ambient air. Gas from the inlet to the analyzer is drawn through the cavity at
a rate of about 100 scm and the conditions in the cavity are held stable at 340 hPa and 45° C.
In its analysis mode the analyzer alternately measures ring-downs in the 7817 cm$^{-1}$ and 7878
cm$^{-1}$ regions. At 7878 cm$^{-1}$ measurements are made at 11 different frequencies, spaced by one
FSR of the cavity and centered at the peak of the Q13Q13 line. Multiple ring-down
measurements are made to improve the precision of the loss determination, with a total of 305
ring-downs allocated to one spectrum.  In the 7817 cm$^{-1}$ region measurements are also made
at 11 distinct frequencies at FSR spacings. Only 35 ring-downs are allocated to this spectral
region, since the measurement of $O_2$ is much more important than water vapor. The data sets
are analysed using a Levenberg-Marquardt fitting routine, which adjusts five free parameters
in each region to find the best agreement to a spectral model based on Galatry line shapes, as
described above. One of the outputs of the 7878 cm$^{-1}$ fit is the frequency offset of the FSR




grid from the center of the Q13Q13 line. This information is used to adjust the position of the
PZT actuated mirror to keep the measurements centered on the line, effectively stabilizing the
optical path length of the cavity to the frequency of the $O_2$ line. The reported water fraction is
obtained by multiplying the fitted amplitude of the water line by a calibration constant derived
from the dilution experiment as explained above. For the $O_2$ fraction a slightly more
complicated procedure is followed. It was observed that the least-squares fitting of the data
gives highly correlated results for the amplitude of the absorption line and the line width
parameter y. The correlation may be due in part to the fitting procedure itself (Press et al.,
1992) and it may also have a contribution from pressure variations that the pressure sensor is
unable to detect. The ratio $A_{O2}/y$ can be determined from the fit much more precisely than
$A_{O2}$ alone and so gives a more sensitive measurement of molecular absorption. It also has the
advantage of being independent of sample pressure, to the extent that the Galatry model
applies (Figure 2). However, using the ratio $A_{O2}/y$ as a metric for absorption adds additional
complications if measurements are to be made over a range of $O_2$ and water concentrations,
because the $O_2/N_2$ ratio and water concentration affect the line width independently of
pressure and $O_2$ concentration alone. To minimize systematic errors due to these broadening
effects, we define a nominal y-parameter based on the measured amplitudes of the $O_2$ and
water lines and the line broadening dependences shown in Figures 3 and 4. The measured
ratio $A_{O2}/y$ is normalized by the nominal y to obtain a quantity that is ideally independent of
pressure and water concentration, and this is the quantity that is multiplied by a calibration
constant to give the reported $O_2$ fraction. In addition, a dry mole fraction is reported for $O_2$,
defined as the directly measured mole fraction corrected for water dilution.
The main goal in developing this instrument was to make high precision
measurements of $O_2$ mole fraction, based on absorption by the dominant $^{16}O_2$ isotopologue.




The absorption lines of the rarer isotopologues are also present nearby, so a mode of operation
was included in which one laser is scanned over neighboring lines of $^{16}O_2$ and $^{16}O^{18}O$ and the
ratio of amplitudes is used to derive an isotopic ratio, reported in the usual delta notation. In
this case the operating pressure was reduced to 160 hPa to improve the resolution of the
nearby lines. The lines measured were the Q3Q3 line of $^{16}O_2$, at 7882.18670 cm$^{-1}$, and the
Q9Q9 line of $^{16}O^{18}O$, at 7882.050155 cm$^{-1}$. The measurement procedure is very much like
that for the $O_2$ fraction measurement, so it will not be described in detail, only the main
differences will be noted. One is that in determining an isotopic ratio there is no advantage to
be obtained from normalizing absorption amplitudes to line widths, instead we simply take
the ratio of amplitudes to compute delta. Although the Q9Q9 line and its neighbor Q8Q8 are
the strongest ones in this band, absorption by $^{16}O^{18}O$ is still very weak, only about 5x10$^{-9}$ cm$^{-1}$
at the line center under the conditions we used. Consequently, the signal-to-noise that can be
achieved with this analyzer is not adequate to determine both the amplitude and the width of
the $^{16}O^{18}O$ line with useful precision, so in the fitting step the y-parameter of the $^{16}O^{18}O$ line
is constrained to be a constant factor times the fitted y-parameter for the $^{16}O_2$ line.
Additionally, because of the weakness of the rare isotopologue absorption, the majority of
ring-downs in each spectrum is devoted to measuring $^{16}O^{18}O$ i.e. 232 ring-downs in each
spectrum versus only 40 for $^{16}O_2$. This implies that the mole fraction measurement in the
isotopic mode is much less precise than when the analyzer measures the Q13Q13 line alone.
**3. Results and Discussions**
**3.1. Laboratory tests at Picarro, Santa Clara**
3.1.1. Temperature and pressure sensitivity

One set of tests was done to determine how well the goal was met of minimizing the

susceptibility of the concentration measurements to uncontrolled noise or drift of the sample





temperature and pressure. For these tests the analyzer sampled dry synthetic air from a tank
and the temperature and pressure setpoints of the cavity were adjusted upward and downward
from the nominal values, to obtain an estimate of the differential response. We express the
sensitivity to experimental conditions in relative form, that is the derivative with respect to
temperature or pressure divided by the signal under nominal conditions.

From these experiments, we determined a temperature sensitivity of $-2.1 \times 10^{-4}$ $K^{-1}$ and

a pressure sensitivity of $+9.8 \times 10^{-6}$ $hPa^{-1}$. The temperature sensitivity is somewhat larger than
expected based on a calculation using Hitran data to estimate the temperature dependences of
all the quantities that go into the measured absorption of the Q13Q13 line. The pressure
sensitivity is strikingly small, indicating a good cancelation of the pressure dependence of
absorption amplitude and line width. Both temperature and pressure sensitivities are small
enough to have a negligible effect on short-term precision of measurements made with the
stabilized ring-down cavity, though long-term drifts in the sensors are always a matter of
concern.
3.1.2. Measurement precision and Drift

Measurement precision was evaluated by analyzing synthetic air containing nominal

atmospheric concentrations of $CO_2$ and $CH_4$ from an aluminum Luxfer cylinder over a period
of several days. The tank, oriented horizontally and thermally insolated (though not
controlled), was connected directly to the instrument (S/N TADS2001) with a 2-stage
regulator and stainless-steel tubing and reducing the flow with an additional orifice to about
55 sccm. For the isotopic mode of operation, the precision of the measurement was also tested
by making repeated measurements from a tank of clean, dry synthetic air.

Figure 8 shows the time series of the precision test data, displaying the reported

oxygen concentration, the height of the oxygen absorption peak, the width of the oxygen



absorption peak and the ambient temperature. The residual drift of the analyzer, although
small, is nevertheless significant given the stringent targets set forth by the WMO-GAW
program. Possible sources of drift include: temperature drifts due to sensor drift or gradients;
pressure errors due to sensor drift; optical artifacts such as parasitic reflections, higher order
cavity mode excitation, and/or loss nonlinearity that can distort the reported oxygen spectrum.
More work is required to identify and eliminate these small drifts.
The Allan standard deviation of the reported $O_2$ fraction is shown in Figure 9. The
ordinate on this plot is the square root of the Allan variance of reported mole fraction, so 1
ppm in these units corresponds to about 5 per meg in the ratio of $O_2/N_2$. The precision of
averaged measurements improves as $\tau^{-1/2}$ for approximately 5000 s and reaches 1 ppm in less
than 10 minutes and remains below 1 ppm for time scales on the order of about 1 hour.
Figure 10 shows the precision of $\delta(^{18}O)$ (uncalibrated) derived from the ratio of lines
measured at 7882 cm$^{-1}$. Because of the weak signal from the $^{16}O^{18}O$ line, it is necessary to
average for more than 20 seconds or more to achieve 1‰ precision on the isotopic ratio. As
for the concentration measurement, averaging improves the measurement precision for times
scale up to about 1 hour.
**3.2. Laboratory measurements at the University of Bern**
3.2.1. Measurements of standard gases
The performance of the instrument was tested by analyzing eight standard gases with
precisely known $CO_2$ and $O_2$ compositions (Table 1) using the CRDS analyzer and comparing
it to parallel measurements with a paramagnetic oxygen sensor (PM1155 oxygen transducer,
Servomex Ltd, UK) embedded to a commercially available Oxzilla fuel cell oxygen analyzer
(OXZILLA II, Sable Systems International, USA) as well as with an isotope ratio mass
spectrometer (IRMS, Finnigan Delta$^{Plus}$XP). The design of the measurement set-up is shown
in Figure 11. Standard gases were directly connected to the pressure controlling unit, and a
multi-port valve (V2) was used to select among the standard gases. The flow from each
cylinder was adjusted to about 120 ml min$^{-1}$ which was eventually directed to a selection
valve (V1), allowing switching between ambient air and standard gases. Flow towards and out
of the Oxzilla was controlled by the pressure controlling unit. The $O_2$ mixing ratio of this
incoming gas was first measured on the Paramagnetic $O_2$ sensor and then directed towards a
non-dispersive infrared analyzer (NDIR) (Li-7000, LICOR, USA) for measuring $CO_2$ and
$H_2O$. The outflow from this analyzer (100 ml min$^{-1}$) returns to the pressure controlling unit
and was eventually divided between the CRDS analyzer (which uses about 75-80 ml min$^{-1}$)
and the IRMS (~ 20 ml min$^{-1}$) via a Tee-junction. Each cylinder was measured for two hours
in each system controlled by a Lab VIEW program.
In priori, we investigated the influence of this Tee-junction, which splits the gas flow
between the CRDS and the IRMS, on the measured $O_2$ values. Manning (2001) showed that
the fractionation of $O_2$ in the presence of a Tee-Junction is strongly dependent on the splitting
ratios as well as temperature and pressure gradients. Hence, we measured and compared the
$O_2$ mixing ratios of two standard gases (CA07045 and CA060943) in two cases: i) in the
presence of a Tee-junction with different CRDS to IRMS splitting ratios and ii) without a
Tee-junction so that all gas flow is directed towards the CRDS analyzer. The splitting ratios in
these test experiments vary from 1:1 to 1:100, and reversed to change the major flow direction
either to the CRDS or the IRMS. Note that the experimental condition in this manuscript is
with a 4:1 splitting ratio (i.e. ~ 80 ml min$^{-1}$ towards the CRDS analyzer and ~ 20 ml min$^{-1}$
towards the IRMS).
In the cases of the smaller splitting ratios (1:1, 1:4 and 4:1), which are relevant for the
results presented in this study, only minor differences in the measured $O_2$ mixing ratios were



observed when compared to case b (i.e. without a Tee-junction). For these two cylinders
measured, the average differences in these cases were about 0.5 ppm, calculated as the mean
of the differences in the raw $O_2$ measurements of the last 60 seconds. The negligible
fractionation can indeed be the result of smaller splitting ratios while strong influence is
usually expected in case of larger splitting ratios (Stephens et al., 2007). For higher splitting
ratios, the result seems inconclusive without any dependence on the ratios due to the strong
decline in the cylinder temperature (specifically at the pressure gauge) caused by higher flow
to achieve the higher splitting ratios (as high as 1:100). Hence, these tests need to be
conducted in a temperature controlled condition and the results could not be discussed in this
manuscript.
Figure 12 shows the standard gas measurements for the seven cylinders with known
$CO_2$ and $O_2$ mixing ratios (Table 1) using both the CRDS and the Paramagnetic analyzers.
Standard eight, which has too high $O_2$, is not shown in the figure as the figure is zoomed-in to
better illustrate the change in $O_2$ for the remaining cylinders. While the first five cylinders
contain $O_2$ and $CO_2$ fractions comparable to ambient air values, standards 6 & 8 had either
very low and very high $O_2$, respectively. In addition, standard 6 and 7 have very low and very
high $CO_2$ mixing ratios. Note that due to its very high $CO_2$ content (~ 2700 ppm), standard 7
was not measured on the IRMS and hence the $O_2$ mixing ratios are unknown. The measured
mixing ratios for the six standard gases between the two systems are in very good agreement
while cylinder 7 showed an opposing signal for the two analyzers compared to standard 6
(Figure 12). While the Paramagnetic analyzer showed a higher $O_2$ mixing ratio, the values
from the CRDS analyzer are lower in $O_2$. This can be associated with the very high $CO_2$
mixing ratio in standard 7, which leads to a strong dilution effect in the CRDS analyzer as it
does not include any correction function for dilution effect from $CO_2$. However, such high



431 $CO_2$ mixing ratios may not be that important for most atmospheric research. Yet, it should be

432 considered to include a parallel $CO_2$ mixing ratios measurement to the instrument as it will

433 further improve the accuracy. This would be especially important for biological or

434 physiological studies where a wide range of $CO_2$ and $O_2$ concentrations must be expected.

435  The measurement precision of the CRDS analyzer was calculated as the standard error

436 of the mean i.e. the standard deviation (1-$\sigma$) of the last 1-minute raw measurements divided

437 by the square root of the number of measurements (n = 60), and for all these cylinders the

438 values are usually between 0.5 ppm to 0.7 ppm. For parallel measurements of these cylinders

439 using a Paramagnetic analyzer, we obtained a precision of about 1 ppm, calculated exactly the

440 same way.

441  We also made a correlation plot to see which of the two instruments are in better

442 agreement with the assigned values based on IRMS measurements for the individual

443 cylinders. While similar correlation coefficients were observed for both analyzers, different

444 slopes were calculated (Fig. A.1). This is due to the fact that the IRMS measures the $O_2$ to $N_2$

445 ratio ($\delta(O_2/N_2)$) in per meg, while the CRDS and the Paramagnetic analyzers provide non-

446 calibrated $O_2$ mixing ratios in units of ppm and per meg, respectively. If we exclude the two

447 standard gases with the highest and lowest $O_2$ mixing ratios (standards 7 and 8) that are

448 subjected to strong dilution effects, both the slope and the $r^2$ values decrease from those

449 shown in Figure A.1. But this decrease is larger in the case of the Paramagnetic

450 measurements, implying a slightly better linearity of the CRDS analyzer.

451 3.2.2. Measurements of ambient air

452  Ambient air measurements were conducted from the roof top of our laboratory at the

453 University of Bern to evaluate the analyzer's performance under atmospheric variability.

454 Ambient air was continuously aspirated from the inlet at the roof of the building at a flow rate





of ~ 250 ml min$^{-1}$ which is then dried using a cooling trap kept at -90 °C towards the
switching valve (V1) and measured in similar way to the standard gases as explained above.
The measurement values obtained here were compared with the parallel measurements by the
Paramagnetic sensor to test the instruments stability and accuracy.

Figures 13 panels a &b show the 1-minute average ambient air measurements from the

rooftop inlet by the Paramagnetic and the CRDS analyzers at the beginning of the testing
period including standard gases measured every 12-hour. While the Paramagnetic analyzer
seems to be stable, the CRDS analyzer showed a strong drift for an extended period. This can
be due to unstable conditions in the CRDS measurement system as it started operating right
after it was unpacked. Hence, we looked into its DAS temperature and pressure records,
which were stable within the manufacturer's recommended range during this period. As the
CRDS analyzer incorporates a water correction function, interference from this species should
be well accounted. Even comparing the analyzer's parallel water measurements to water
measurements by the NDIR system such a drift was not observed. It should be noted that the
two internal standard gases which were less frequently measured (every 12 hours) during this
period were also drifting in similar pattern. This implies that the drift is associated with the
analyzer. Interestingly, we observed that the two cylinders follow exactly the same drift
pattern that can be modeled using a polynomial function which can then be used to correct for
the observed drift in the ambient air measurements. After applying a polynomial drift
correction, we were able to fully account for the observed drift. However, the manufacturer
decided to further investigate possible causes of this drift. After further improvements, we
obtained the first commercial analyzer in September 2017 and repeated the above tests
(Figure 13 c &d). No such drift was observed any more in the standard gases or in ambient air
measurements.



3.2.3. Water correction test
Measurements of oxygen are reported as both wet ($O_{2,\ raw}$) and dry ($O_{2,\ dry}$) mole
fractions by the CRDS analyzer as it also measures water vapor in parallel at its water
absorption line (7817 cm$^{-1}$), and corrects for the dilution effect based on an inbuilt numerical
function:
$$O_{2,dry} = \frac{O_{2,raw}}{1 - f_{H_2O}} \qquad\qquad (4)$$
where $f_{H2O}$ is the measured water mole fraction.
The efficiency of water correction by this function was assessed in two ways: i) by comparing
the water vapor content in standard air measured by this analyzer with similar measurements
by the NDIR analyzer and ii) by comparing the oxygen mixing ratios between non-dried
ambient air measured and corrected for water dilution by the CRDS analyzer with dried air
measured using a paramagnetic analyzer.
Figure 14 shows the water vapor content for standard gases measured continuously for
two days by the CRDS and the NDIR analyzers. Note that the two data sets are manually
fitted to each other as the measured water values by the NDIR analyzer are not calibrated.
Based on these plots, the two analyzers are in very good agreement although there are small
differences during very dry conditions (low water content).
The water correction test was conducted by measuring dried ambient air (Figure 15a)
into both analyzers as well as allowing non-dried air to the CRDS analyzer only (Figure 15b)
and comparing the difference in $O_2$ measurements in both cases (Figures 15c & 15d). shows
the water contents of dried ambient air measured in both analyzers while Figure 15b shows in
case non-dried air is admitted to the picaro analyzer only (note that the CRDS uses its in-built
water correction function). The measurements of the Paramagnetic analyzer were scaled to





ppm units by applying the correlation equation obtained from the six standard gas
measurements of the two analyzers (Fig. A.1). Note that the CRDS measurements were
corrected for the observed drift using the polynomial fit to the two standard gas measurements
stated above.

In the first period of the measurement when both analyzers measured dried ambient

air, the absolute differences between the 1-minute averages measured over two days by the
two analyzers were mostly within 15 ppm and symmetrically distributed around zero.
However, when wet air was admitted to the CRDS analyzer and the in-built water correction
was applied, a stronger variability was observed in the calculated differences. This implies
stronger short term variability in the CRDS analyzer measurement values (as nothing was
changed for the Paramagnetic measurement system) when wet samples were analyzed. The
more negative values in the differences can also be associated with overestimation of the $O_2$
mixing ratios by the CRDS originating from an overestimated water correction. However,
detailed evaluation of the analyzer's water correction function is beyond the scope of this
study.
**3.3. Field Measurements**

After a series of tests at University of Bern, we conducted multiple field measurements

at the High Altitude Research Station Jungfraujoch and the Beromünster tall tower sites in
Switzerland described below.
3.3.1. Tests at the High Altitude Research station Jungfraujoch

The High Alpine research station Jungfraujoch is located on the northern ridge of the

Swiss Alps (46° 33′ N, 7° 59′ E) at an elevation of 3580 m a.s.l. It is one of the global
atmospheric watch (GAW) stations well-equipped for measurements of numerous species and
aerosols. The site is above the planetary boundary layer most of the time due to its high


elevation (Henne et al., 2010; Zellweger et al., 2003). However, thermally uplifted air from
the surrounding valleys during hot summer days or polluted air from the heavily industrialized
northern Italy may reach at this site (Zellweger et al., 2003). The Division of Climate and
Environmental Physics at the University of Bern has been monitoring $CO_2$ and $O_2$ mixing
ratios at this site based on weekly flask sampling and continuous measurements since 2000
and 2004, respectively. The $CO_2$ mixing ratio is measured using a commercial NDIR analyzer
(S710 UNOR, SICK MAIHAK) while $O_2$ is measured using the Paramagnetic sensor
(PM1155 oxygen transducer, Servomex Ltd, UK) and fuel cells (Max-250, Maxtec, USA)
embedded within a home-built controlling unit. Similar to the comparison tests at the
University of Bern, we have conducted parallel measurements between the CRDS analyzer
and the paramagnetic cell at this high altitude site during 03 – 14 February 2017. The
measurement of ambient air at the Jungfraujoch system is composed of sequential switching
between a low span (LS) and high span (HS) calibration gases followed by a target gas (T)
measurement (once a day) to evaluate the overall system performance and finally a working
gas (WG) measurement before switching back to ambient air.

Figure 16 (top panel) shows the calibrated 1-minute averaged $O_2$ mixing ratios

measured at this high altitude site in comparison with the Paramagnetic oxygen analyzer
already available at the site. While a strong variability was observed during the measurement
period of 10-days by both analyzers, a very good agreement was observed between them.

Figure 16 (bottom panel) shows the absolute difference of 1-minute averages in

atmospheric $O_2$ measured at Jungfraujoch between the two analyzers which are mostly within
±5 ppm range (but sometimes going as high as ±10 ppm) without an offset. However, for
generally reported 10-minutes, half-hourly or hourly means these values correspond to < 1.5
ppm, < 1 ppm and < 0.65 ppm.





3.3.2. Tests at the Beromünster tall tower site

The Beromünster tower is located near the southern border of the Swiss Plateau, the comparatively flat part of Switzerland between the Alps in the south and the Jura mountains in the northwest (47° 11′ 23″ N, 8° 10' 32″ E, 797 m a.s.l.), which is characterized by intense agriculture and rather high population density. A detailed description of the tower measurement system as well as a characterization of the site with respect to local meteorological conditions, seasonal and diurnal variations of greenhouse gases, and regional representativeness can be obtained from previous publications (Berhanu et al., 2016; Berhanu et al., 2017; Oney et al., 2015; Satar et al., 2016). The tower is 217.5 m tall with access to five sampling heights (12.5 m, 44.6 m, 71.5 m, 131.6 m, 212.5 m) for measuring CO, $CO_2$, $CH_4$ and $H_2O$ using Cavity Ring Down Spectroscopy (Picarro Inc., G-2401). By sequentially switching from the highest to the lowest level, mixing ratios of these trace gases were recorded continuously for three minutes per height, but only the last 60 seconds were retained for data analysis. The calibration procedure for ambient air includes measurements of reference gases with high and low mixing ratios traceable to international standards (WMO-X2007 for $CO_2$ and WMO-X2004 for CO and $CH_4$), as well as target gas and more frequent working gas determinations to ensure the quality of the measurement system. From two years of data a long-term reproducibility of 2.79 ppb, 0.05 ppm, and 0.29 ppb for CO, $CO_2$ and $CH_4$, respectively was determined for this system (Berhanu et al., 2016).

Between 15.02.2017 and 02.03.2017, we have connected the new CRDS oxygen analyzer in series with the $CO_2$ analyzer (Picarro G-2401) and measured the $O_2$ mixing ratios at the corresponding heights. Similar to the $CO_2$ measurements, $O_2$ was also measured for three minutes at each height. During this period, we have evaluated the two features (isotopic mode and concentration mode) of the CRDS analyzer. In the isotopic mode, the CRDS

measures the $\delta^{18}O$ values as well as the $O_2$ concentration while in concentration mode only
the latter was measured.

During the tests conducted at this tower site, we first evaluated the two operational

modes (concentration vs isotopic modes) of the CRDS analyzer. Ambient air measurements
on isotopic mode over a 4-days period showed a strong variability in the measured oxygen
mixing ratios and it was not possible to distinguish the variability in the $O_2$ mixing ratios
among the five height levels. The calculated 1-minute standard error for ambient air
measurements was as high as 10 ppm while a standard error of less than 1 ppm was
determined from similar measurements in the concentration mode. Additionally, comparing
the $O_2$ values between the two modes, frequent short time variation in ambient air $O_2$ (~ 200
ppm) was observed in the isotope mode measurements while the variation in the concentration
mode is significantly smaller (~ 30 ppm). This precision degradation is due to the weaker $^{16}O$
oxygen line used for the isotopic mode, and the fact that far more ring-downs are collected on
the rare isotopologue in isotopic mode Hence, we have conducted the remaining test
measurements in concentration mode.

As this tower has five sampling height levels, we first followed three minutes of

switching per inlet level, which enables four measurements per hour at a given level.
However, we noticed hardly any difference among the different levels due to strong short
term variability in $O_2$ mixing ratios between the consecutive heights. Hence, we switched to a
longer sampling period of six-minutes per height. Figure 17 shows the diurnal $CO_2$ and $O_2$
variations at the lowest (12 m) and highest (212.5 m) sampling heights of the tower. These
two heights were selected simply to better illustrate the difference in the mixing ratios. The
$CO_2$ mixing ratios on the top panel show higher values at the 12 m inlet than the highest level
most of the day due to its closeness to sources except during the afternoon (11:00 - 17:00


UTC) when both levels show similar but decreasing $CO_2$ mixing ratios. This is due to
presence of a well-mixed planetary boundary layer (PBL) (Satar et al., 2016). The lag in $CO_2$
peak between the two height levels by about two hours indicates the duration for uniform
vertical mixing along the tower during winter 2017. The opposite variability patterns are also
clearly visible in the $O_2$ mixing ratios shown in the lower panel with a clear distinction
between the two height levels during early in the morning and in the evening while similar $O_2$
values were observed in the afternoon. These opposing profiles are expected as $CO_2$ and $O_2$
are linearly coupled with a mean oxidation ratio of -1.1 ± 0.05 (Severinghaus, 1995) for land-
biospheric processes (photosynthesis and respiration) and -1.44 ± 0.03 for fossil fuel burning
(Keeling, 1988b).
Table 2 shows the oxidation ratios derived as the slopes of the linear regression
between $CO_2$ and $O_2$ mixing ratios at the different height levels measured on 25 February
2017. Accordingly, height dependent slopes were observed with a slope of -0.98 ± 0.06 at the
lowest level, close to the biological processes induced slope but slightly lower than its mean
value. For the highest level, we calculated a slope of -1.60 ± 0.07 a value close to fossil fuel
combustion oxidation ratio. Note that depending on fossil fuel type the oxidation ratio can
range between -1.17 and -1.95 for coal and natural gas, respectively (Keeling, 1988b). While
the slopes derived for the two other levels (44.6 m and 131.6 m) show similar values between
the highest and lowest height levels, possibly from mixed sources, the middle level showed a
slightly higher slope than these two levels but still in the large range between the lowest and
highest inlet heights.
3.4. Evaluation of the $\delta^{18}O$ measurements
To further evaluate the analyzer's performance in measuring stable oxygen isotopes,
we conducted ambient air isotopic composition measurements as well as analyzed a standard





gas without $CO_2$ which has a known $\delta^{18}O$ value. The choice of this $CO_2$-free air standard gas
is twofold: one it has a known $\delta^{18}O$ value and second as it has no $CO_2$ possible interference
from band overlap is avoided. For this test three 0.5 L glass flasks were preconditioned and
filled with this standard gas to ambient pressure. These flasks were attached before or after
the water trap (Fig. 11) and measured similar to ambient air measurements. These
measurements were then compared with $\delta(^{34}O/^{32}O)$ values obtained by parallel measurements
using our IRMS.

Figure 18 shows the $\delta^{18}O$ values of ambient air from the roof top with three

consecutive measurements of glass flasks filled with $CO_2$-free air in-between followed by a
fourth flask filled with breath air. An excellent agreement was observed for measurements
from both instruments for the three flasks filled with a standard gas. However, the fourth flask
with breath air showed a signal opposite to the measurements by the IRMS. As breath air
contains large amount of water $CO_2$ in addition to $O_2$, which can possibly interfere with the
CRDS analyzer measurements, we have removed $H_2O$ and $CO_2$ by using a cryogenic trap (-
130 °C) and in an additional experiment using Schütze reagent to remove both CO and $CO_2$.
However, we have not observed any improvement towards an agreement with the IRMS
measurements. Therefore, any other gas component in the breath air must be relevant for the
interference. Based on the absorption lines in the spectral range of the instrument (7878 cm$^{-1}$)
retrieved from HITRAN database, we expect interference either from carbon monoxide (now
excluded by the tests) or methane or VOCs including acetone, ethanol, methanol or isoprenes,
all of which have been measured in breath air (Gao et al., 2017; Gottlieb et al., 2017; Mckay
et al., 1985; Ryter and Choi, 2013; Wolf et al., 2017). Further investigations have to shed light
on these interferences in order to take corresponding action to surpass these shortcomings in
the isotope analysis based on cavity ring-down spectroscopy.



## 4. Conclusions

We have thoroughly evaluated the performance of a new CRDS analyzer which measures $O_2$ mixing ratios and isotopic composition combining laboratory and field tests. Even if a drift in the analyzer was observed at the beginning of this study, which if it appears can be easily corrected by calibration, the recent analyzers built by the manufacturer did not show such instrumental drift. However, prior tests are recommended to see the analyzer's stability.

The T-split tests for the current measurement setup based on the measurements of two standard gases showed a difference within the measurement uncertainty. However, this effect may become significant while applying larger splitting ratios and we recommend conducting further experiments to accurately quantify this influence for larger splitting ratios.

We have observed a strong influence of dilution in the measured $O_2$ values during the presence of high $CO_2$ mixing ratios. Even if such an influence may not be critical for the present study, such an effect might be significant in other studies where higher $CO_2$ mixing ratios might be present and we recommend following a correction strategy based on parallel $CO_2$ measurements. This also applies for more accurate analysis.

The water correction applied by the instrument's in-built function seems to sufficiently correct for the water vapor influence. However, a larger variability of the difference was observed between the CRDS analyzer and the Paramagnetic cell when dried samples were used in both systems. This can possibly be due to an overcorrection by the water correction function of the CRDS analyzer when dried samples were used. This is particularly true for the very low water vapor range ($< 100$ ppm).

Based on the analysis of $O_2$ mixing ratios in the concentration and isotopic modes, we have observed about a significant decrease in precision (about ten-fold) in the latter



measurement mode. The measured $\delta^{18}O$ values for the standard air by the CRDS analyzer are
in excellent agreement with the IRMS values. However, such measurements for a breath air
showed a contrasting signal, possibly due to interference from other gases as breath air
contains $CO_2$, $CH_4$ and CO in addition to oxygen. Hence, we recommend further investigation
on such possible contaminants and how to possibly remove them while conducting ambient
air measurements.
**Acknowledgement**
We would like to thank ICOS-RI and the Swiss National Science Foundation (SNF) for
funding  ICOS-CH  (20FI21_148994,  20FI21_148992).  We  are  also  grateful  to  the
International Foundation High Alpine Research Stations Jungfraujoch and Gornergrat. The
measurement system at the Beromünster tower was built and maintained by the CarboCount-
CH (CRSII2_136273) and IsoCEP (200020_172550) projects both funded by SNF.
















List of Tables
Table 1. Assigned mixing ratios of standard gases used in this study and their corresponding
values measured by the NDIR, CRDS and IRMS at the University of Bern. [1]The assigned
values are based on measurements from different institutions (University of Bern (UB),
Scripps or NOAA, see column cylinder name). [2]Measurements are on the Bern scale for $CO_2$
and $O_2$. The Bern scale is shifted by +550 per meg. [3]Values on the Scripps scale.

| Cylinder name | Assigned $CO_2$ (ppm) [1] | Assigned $O_2$ (per meg) [1] | $CO_2$-IRMS (ppm) [2] | $CO_2$-NDIR (ppm) [2] | $O_2$-IRMS (per meg) [2] | $O_2$-Paramagnetic (per meg) [2] | $O_2$-CRDS (per meg)[2] |
|---|---|---|---|---|---|---|---|
| ST-1 LUX3576-UB | 427.47 | -1026 | 427.47 | 427.59 | -1026 | -1070 | -1057 |
| ST-2 LK922131-UB | 368.09 | 599 | 368.09 | 367.82 | 599 | 560 | 590 |
| ST-3 CA07045-Scripps | 382.303 | -271.6 | 382.50 | 381.99 | 278 (-272.2)[3] | 302 | 281 |
| ST-4 CA07043-Scripps | 390.528 | -476.4 | 390.69 | 390.15 | 71 (-479.5)[3] | 66 | 63 |
| ST-5 CA07047-Scripps | 374.480 | -807.7 | 374.70 | 374.17 | -253 (-803.3)[3] | -212 | -233 |
| ST-6 CA04556-NOAA | 192.44 | -3410 | 191.21 | 191.64 | -3410 | -2905 | -3013 |
| ST-7 CA06943- | 2699.45 | - | | 2612.80 | - | -2691 | -3369 |





| NOAA | | | | | | | |
|---|---|---|---|---|---|---|---|
| ST-8 LK76852-UB | 411.49 | 37794 | 411.49 | 406.25 | 37794 | 34513 | 36017 |



Table 2. The $CO_2$ and $O_2$ correlation coefficients at the different height levels derived using
the least square fit and the correlation coefficients ($r^2$). Uncertainties are calculated as
standard error of the slope.

| Height | Oxidation Ratios (O₂:CO₂) |
|---|---|
| 12.5 m | -0.98 ± 0.06 (0.48) |
| 44.6 m | -1.29 ± 0.07 (0.50) |
| 71.5 m | -1.49 ± 0.08 (0.47) |
| 131.6 m | -1.23 ± 0.05 (0.55) |
| 212.5 m | -1.60 ± 0.07 (0.61) |













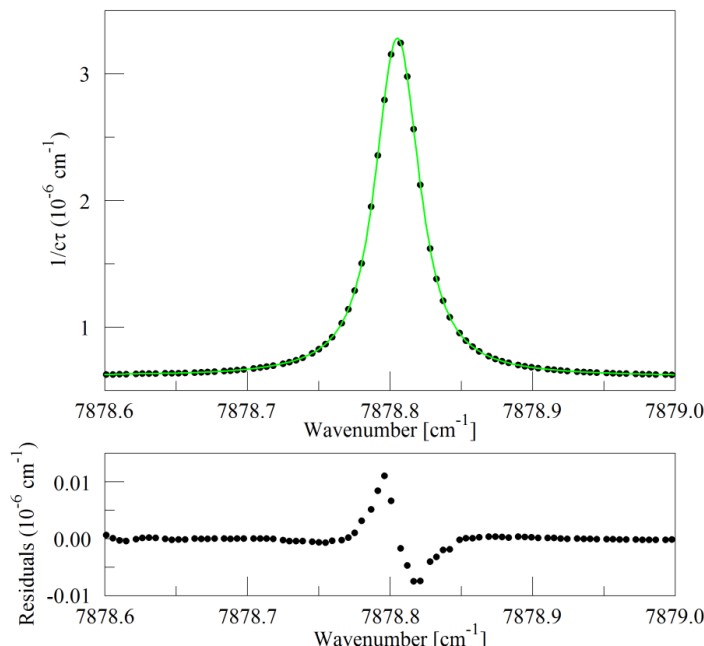

Figure 1. The Q13Q13 line of $O_2$ measured in a sample of synthetic air at a sample
temperature and pressure of 45° C and 333 hPa, respectively.












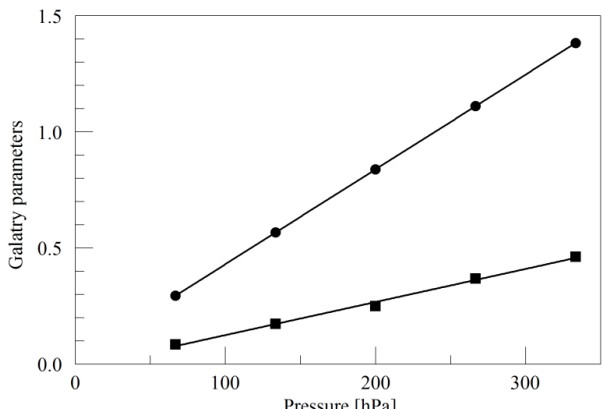


Figure 2. Best-fit values for the Galatry parameters of the Q13Q13 line of $O_2$, as a function of
pressure. The line broadening parameter y is represented by circles and the line narrowing
parameter z by squares. The solid lines are linear fits to the measurements. The best-fit offset
and slope are 0.0227 and 0.004082 $hPa^{-1}$ for y, and -0.0169 and 0.001424 $hPa^{-1}$ for z.





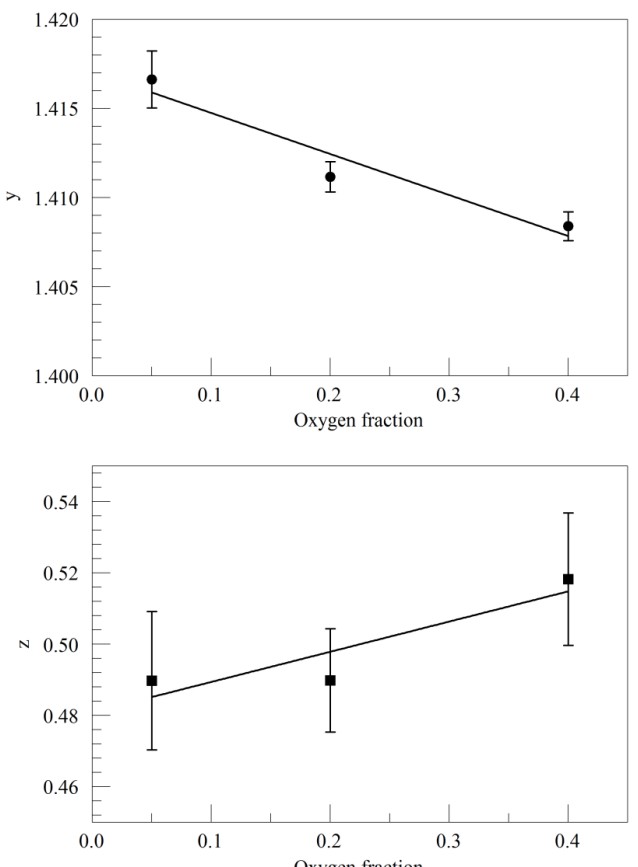


Figure 3. Galatry parameters of the Q13Q13 line of $O_2$ at 340 hPa and 45° C as a function of

O₂ mole fraction in binary $O_2$ - $N_2$ mixtures.

The linear fits to the data are $y = 1.417 - 0.023 \times f_{O2}$ and $z = 0.481 + 0.085 \times f_{O2}$.





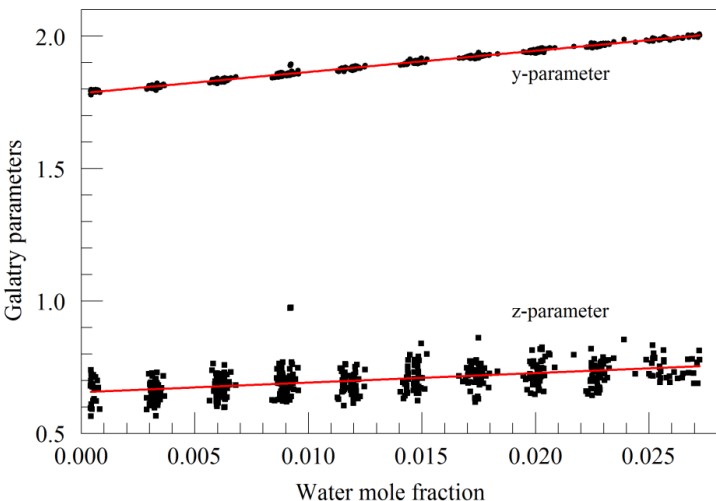


Figure 4. Galatry parameters of the 7816.75210 cm$^{-1}$ water line in air at 340 hPa and 45° C as
a function of water mole fraction. Black points are from measurements and red lines are
linear fits: $y = 1.7846 + 8.01 \times f_{H2O}$ and $z = 0.656 + 3.60 \times f_{H2O}$.





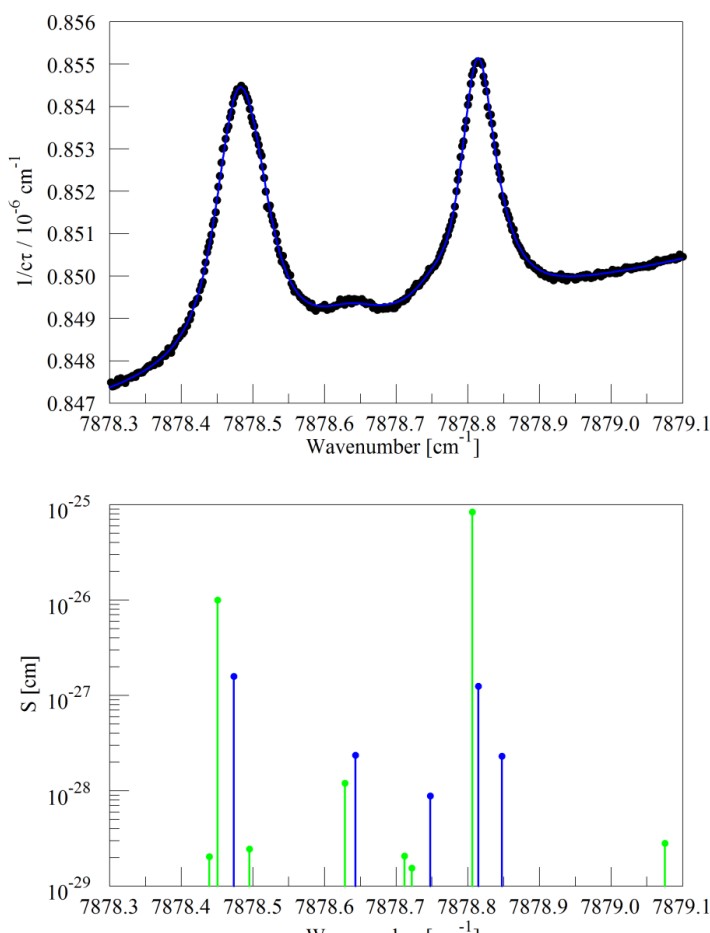


Figure 5. Upper panel: spectrum of water in nitrogen (points) and fit to Voigt model (blue

curve). Lower panel: Oxygen (green) and water (blue) lines in the Hitran database.




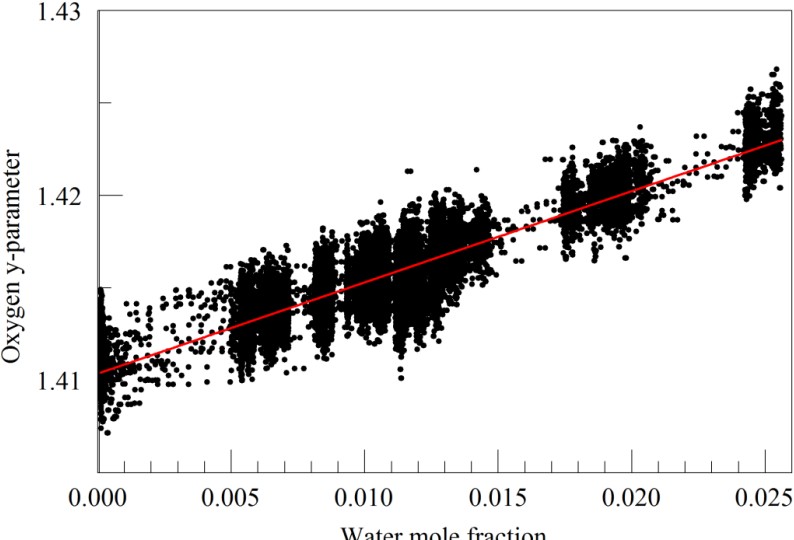


Figure 6. Galatry collisional broadening parameter of the oxygen Q13Q13 line at 340 hPa


and 45° C versus water mole fraction. Black points are from measurements and the red line is
a linear fit: $y = 1.4109 + 0.467\ f_{H2O}$.






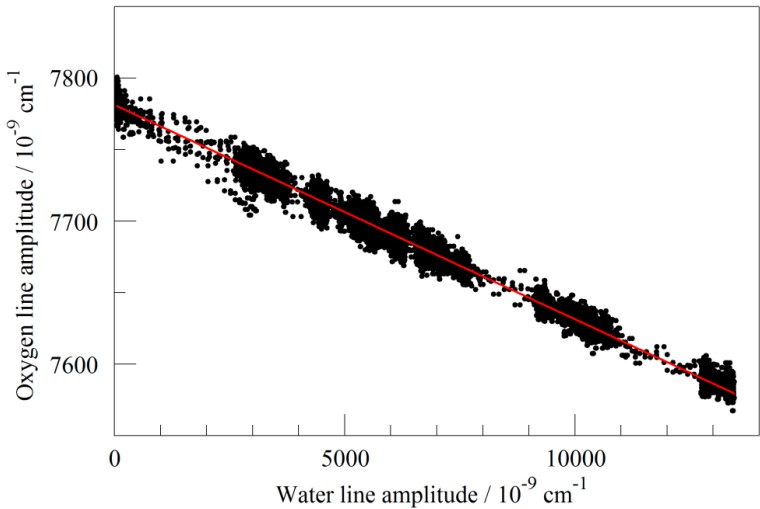



Figure 7.  Measured absorption line amplitudes for oxygen and water vapor for water vapor
mixing ratios ranging from nearly 0 to 0.025.  Black points are from measurements and the
red line is a linear fit: with intercept 7.78001 x $10^{-6}$ cm$^{-1}$ and slope -0.014807.





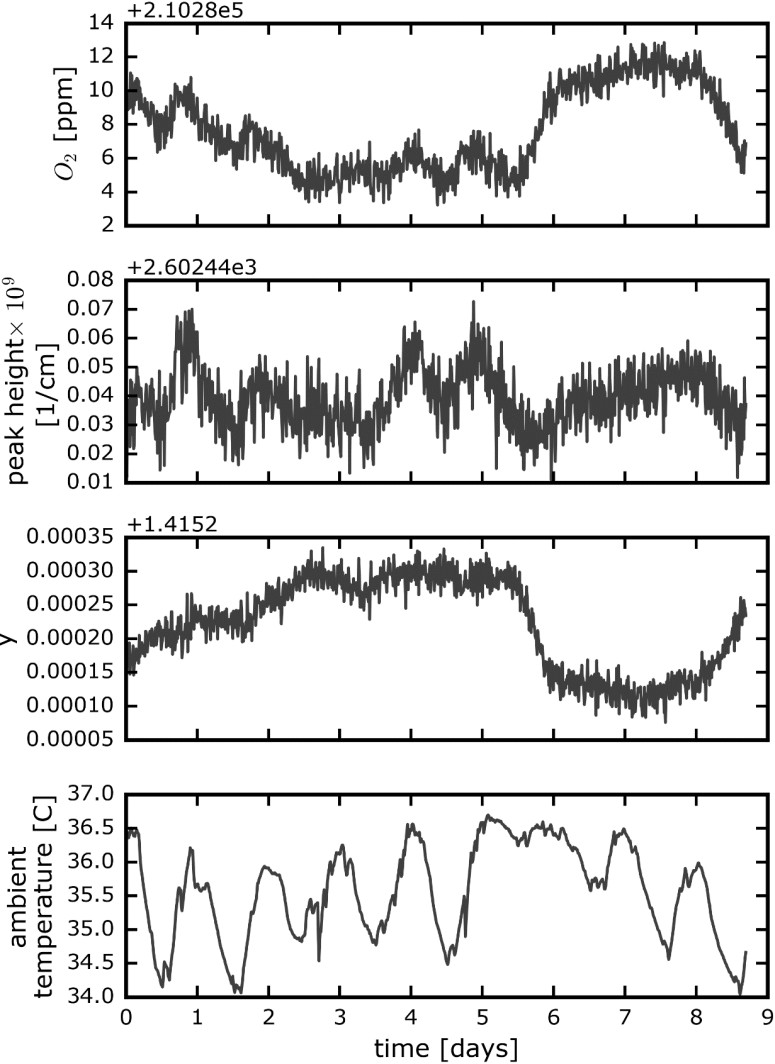


Figure 8. Time series from a measurement of a single tank over about a week. The four panels
shown the water-corrected oxygen concentration, the absorption peak loss minus the baseline
loss, the measured Lorentzian broadening factor, and the ambient temperature (measured in
the instrument housing), respectively. A windowed average of 300 seconds was applied to all
four data sets.





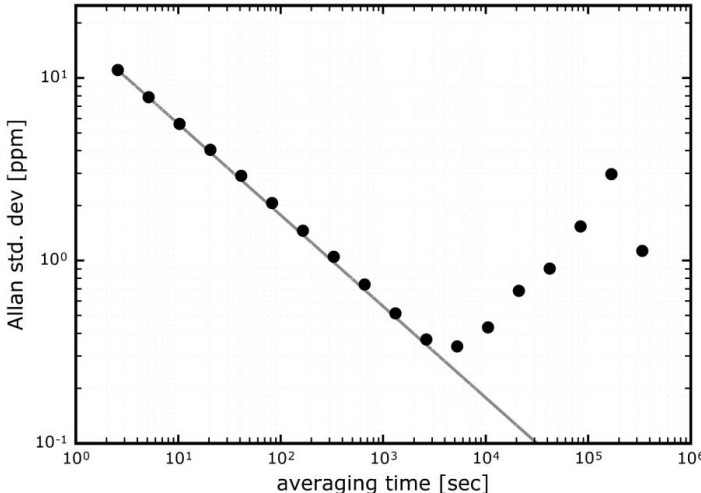


Figure 9. Precision of $O_2$ mole fraction measured from a tank of synthetic air. Filled circles
are measurements and the line shows the ideal $\tau^{-1/2}$ dependence.









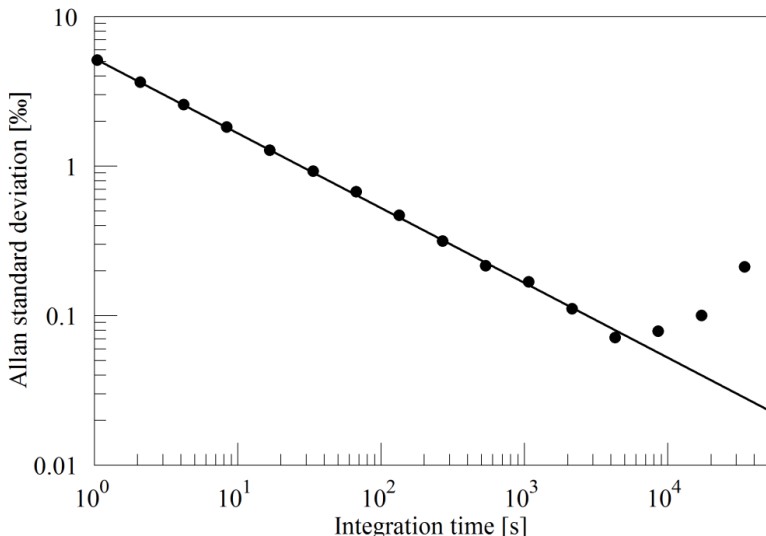


Figure 10. Precision of δ($^{18}$O) measured from a tank of synthetic air. Filled circles are
measurements and the line shows the ideal $\tau^{-1/2}$ dependence.










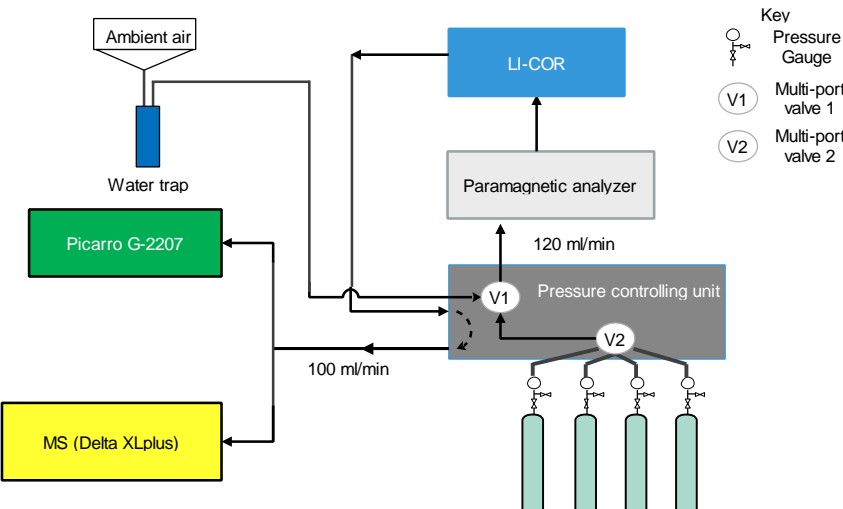


Figure 11. Schematics of the measurement system used to compare the Picarro analyzer with
the Mass Spectrometer at Bern.













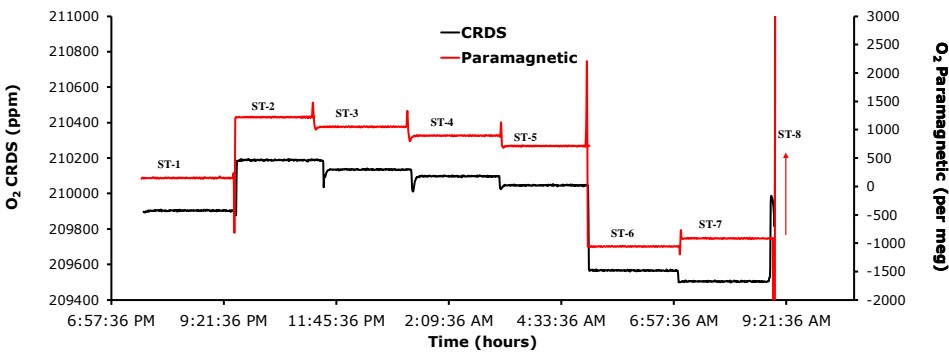


Figure 12. Comparison of oxygen mixing ratios for the seven standard gases measured using
the CRDS analyzer (black) and the Paramagnetic sensors (red).




















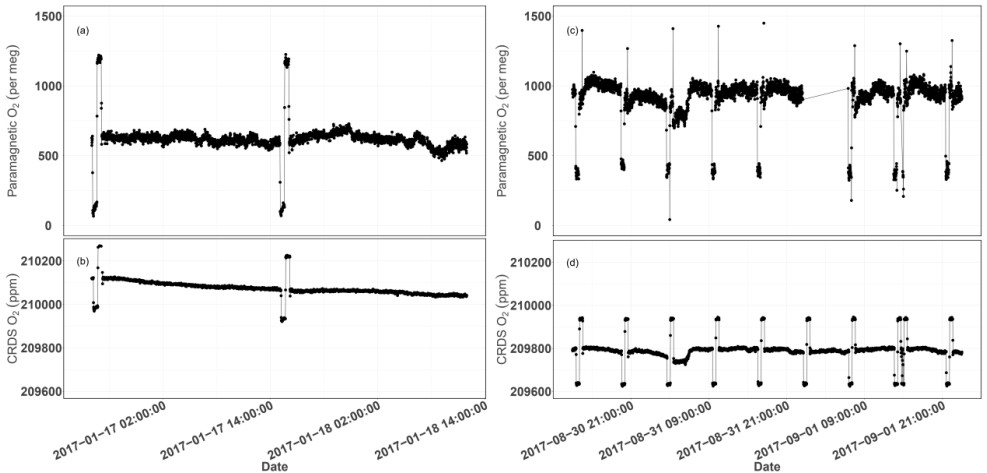


Figure 13. Parallel ambient air measurements by the Paramagnetic and CRDS analyzers at the
beginning of the testing period (Panels a & b, January 2017) and the second phase of testing
(Panels c & d, September 2017). The spikes are measurements from the two standard gases
bracketing the ambient air values.














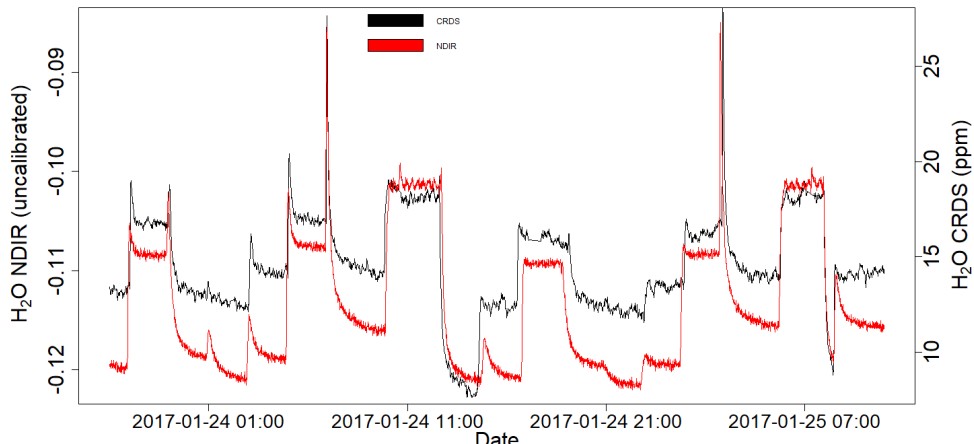


Figure 14. Parallel water vapor measurements for a dried ambient air by both the NDIR and

CRDS analyzers. Note that the water values from the NDIR analyzer are not calibrated.


















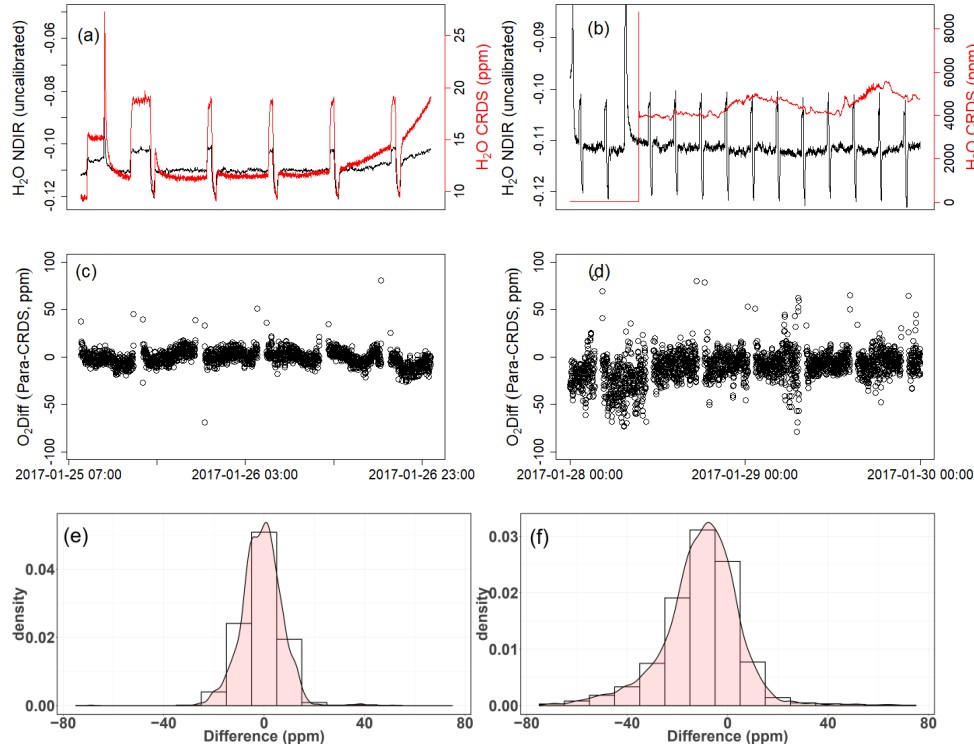

Figure 15. Results of water correction tests. Water measurements of the NDIR (left scale) for
dry conditions (a,b) and the CRDS analyzer (right scale) for dry (a) and wet (b) conditions.
The difference in oxygen measurements between the Paramagnetic and the CRDS instrument
using the built-in water correction for the CRDS values under dry (c) and wet (d) conditions.
Panels (e) and (f) show the population density functions.







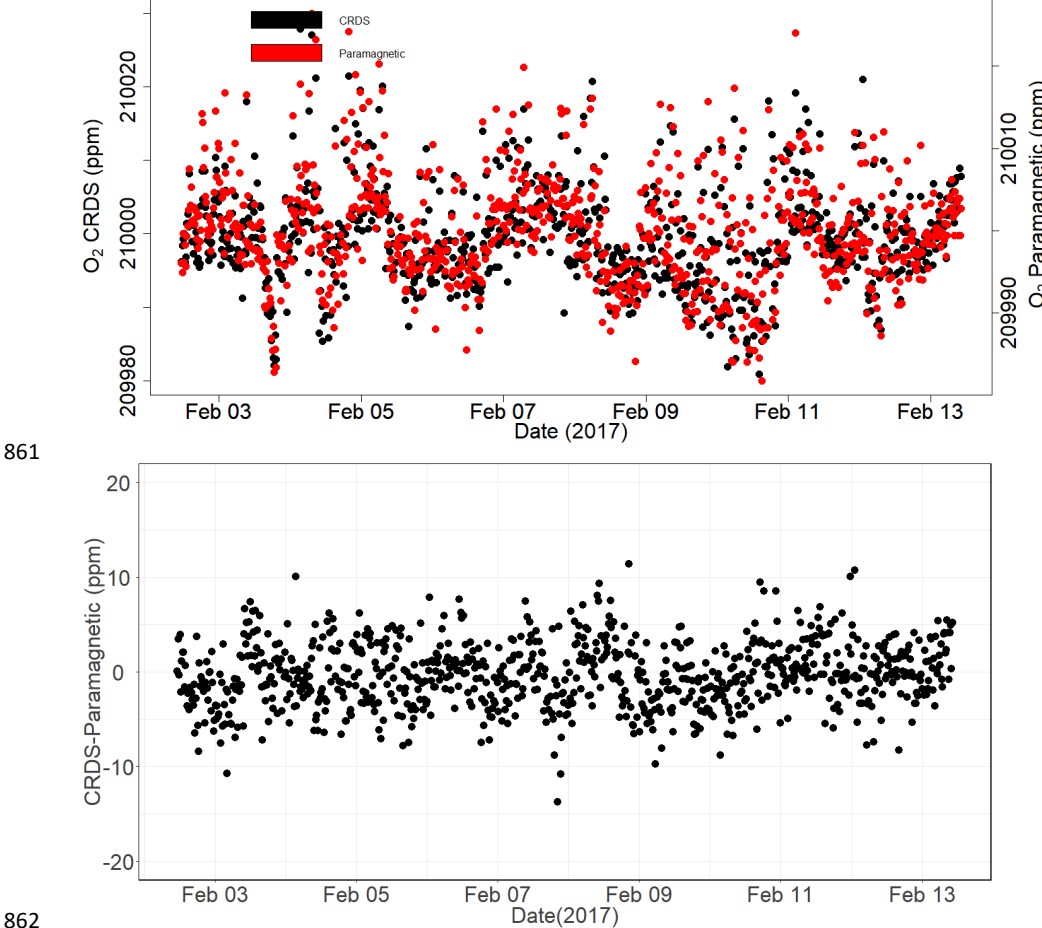



Figure 16. Calibrated ambient air oxygen measurements (1-minute average) at the Jungfraujoch site using the CRDS and Paramagnetic analyzers both in ppm units (a) and the absolute difference between the two measurements in ppm (b) by matching time stamps.












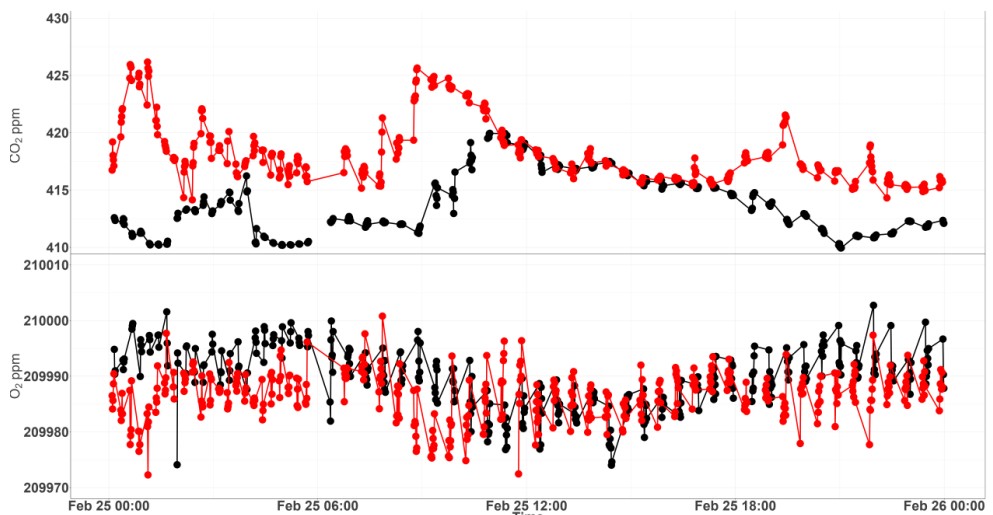

Figure 17. Diurnal variations of $CO_2$ (top) and $O_2$ (bottom) measurements from the 12 m (red)

and the 212.5 m (black) height levels at Beromünster tower.





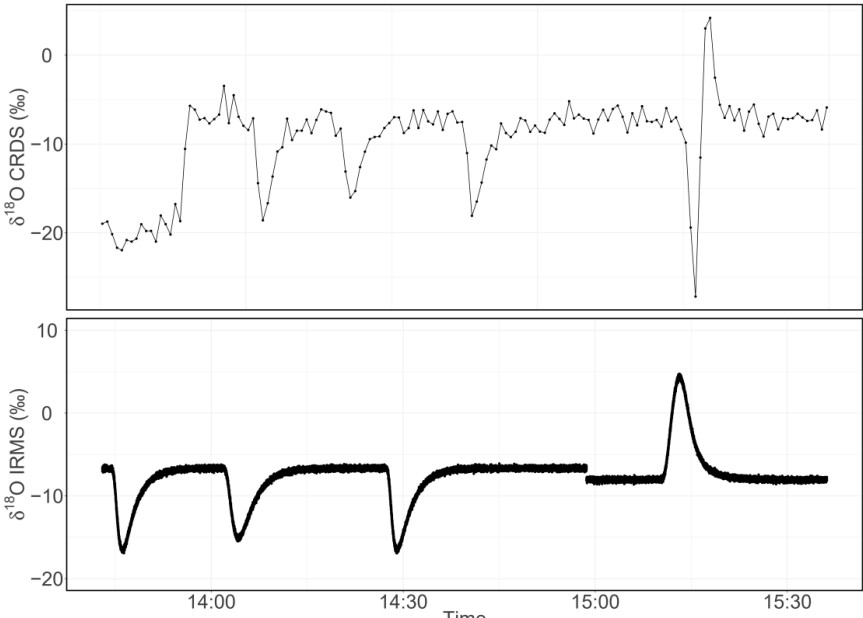


Figure 18. Consecutive $\delta^{18}O$ measurements of a standard gas ($CO_2$-free air) filled into three

flasks followed by measurement of breath air using the CRDS analyzer (top) and IRMS

(bottom). These measurements were carried out in the middle of ambient air measurements.















**Appendix A.**
**Additional plots**

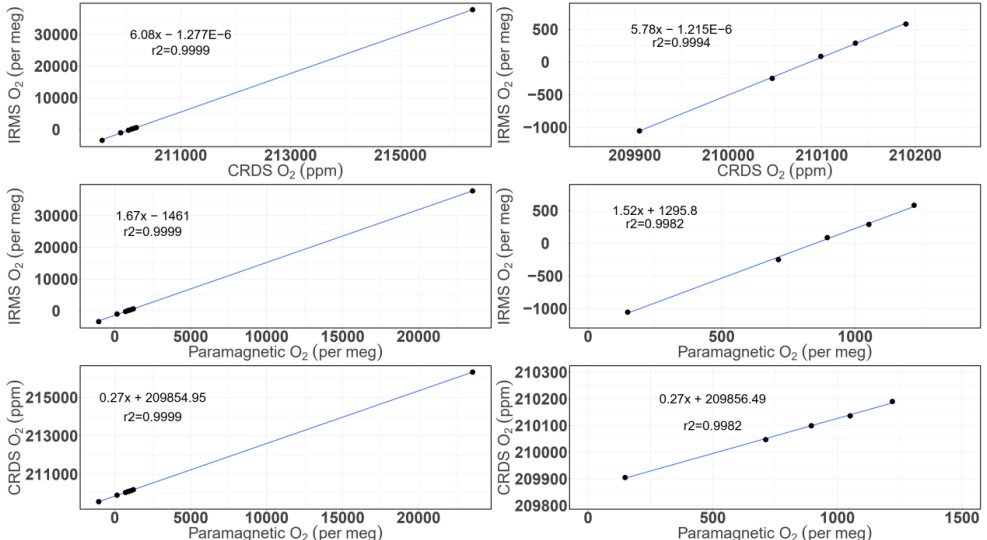


Figure A.1. Correlations between the $O_2$ mixing ratios measured by the CRDS and
Paramagnetic analyzers with the mass spectrometric measurements (uncalibrated values). The
left panels are for all the cylinders measured (standards 1 to 8) while the right ones are after
zooming only to standards 1-5.









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
