# Peer review of "High-precision atmospheric oxygen measurement comparisons between a newly built"

_Atmospheric Measurement Techniques, 2018_

## Referee Comment (RC1) · Anonymous Referee #1 · 29 Jan 2019

General comments

The paper entitled *High-precision atmospheric oxygen measurement comparisons between a newly built CRDS analyzer (Picarro G-2207) and existing measurement techniques* describes a Cavity Ring Down Spectrometer devoted to the determination of oxygen concentration in air and to the delta $^{18}O$ isotopic ratio measurement depending of the instrument mode. The performances of the instrument are tested in laboratory by comparing measurements of well-known samples with results obtained using other techniques (IRMS, paramagnetic technique, Licor instrument) on the same samples. In the field measurements and comparisons are also provided by the authors at the High Altitude Research Station Jungfraujoch and at the Beromünster tall tower. The paper is well-written and detailed. The paper is now much easier to read thanks to the new way the experimental description and the obtained results are presented. The performances reached by the CRDS instrument are at the state-of-the-art for optical methods. The paper is fully in the scope of AMTD and is well-suited for a publication in this journal but needs some significant corrections (see the comments below).

Main remarks:

Part of the work in Fleisher et al. 2015 to which the authors refer in lines 135-141 seems to not be published. If authors have a published reference corresponding to Fleisher et al. 2015 they can let the text as it is but if they are not able to give a published reference they have to remove the lines 135-141 as well as the lines 144-146 and the reference.

The concentrations reported in Figures 8, 12, 13 have to be given in the same unit (per meg or better in ppm) to facilitate the comparison. For example in Figure 13 it will be better to plot the two layers (the paramagnetic and the CRDS measurements) on the same graph using the same units.

Specific comments

P2, L33: The authors have to recall the definition of *per meg* unit.

P2, L43:The authors have to recall the definition of *the oxidation ratio* and give some explanations.

P4, L86: The authors have to specify the reference of the pressure gauge, the proportional valve and acquisition board allowing stabilizing the pressure at the level of $3 \times 10^{-5}$!

P4, L91: The authors should give the typical ring-down time they have.

P5, L104: Give here the FSR value instead of p7.

P6, L125: The authors should cite the following reference: Tran et al., JQSRT (2019) 222-223, p108-114.

P6, L142-144: This sentence is strangely written and should be reformulated.

P7, L146-149: The excess noise observed by the authors is probably due to the fact that shorter ring-downs have less data points available for the fit. This should give a noise level increased by a factor of $\tau^{(-3/2)}$ where $\tau$ corresponds to the RD time.

P7, L158: Put also on Figure 1 the residuals when a Voigt profile is used. Moreover, the residuals observed on this figure seem to be due to a frequency shift and not to the limitations of the Galatry model. In the description of the Galatry profile, the authors don't mention the collisional line-shift parameter. Is this parameter taken into account for the calibration procedure? What about line-mixing effect?

P7, L166: *frequency of narrowing collisions* should be replaced by *velocity change collision rate*.

P7, L167: The authors should be more precise: …$\sigma_D$ is the **1/e** Doppler **half**-width of the transition…

P8, L169-170: The units of $k_B$, $T$, $M$ and $c$ have to be given.

P8, L176: Cite HITRAN2016 instead of HITRAN2013.

P8, L177: Give the uncertainty reported in HITRAN data base.

P8, L178: Change the reference if required (see my comment above).

P10, L217-219: Not true for the line near 7878.45 cm$^{-1}$ where the water transition dominates.

P10, L219-220: Three lines of HDO are missing in the figure but are present in HITRAN2016.

P10, L222-L224: This is not true as HDO lines are present.

P11, L245: *the z-parameter was constrained to be proportional to y, based on earlier measurements.* This is not clear for me. What are these earlier measurements and how they show that? Same remark for line 250.

P11, L251-253: How the missing HDO lines will be treated as the HDO isotopic abundance is not determined from the 7817 cm$^{-1}$ window.

P12, L277: *sc**c**m* instead of *scm*.

P13, L299-308: Maybe adding equations will make this paragraph easier to understand.

P14, L326-329: What about the water lines in that spectral region (especially the $H_2O$ line near 7881.98 cm$^{-1}$)?

P19, L436: It is strange that only the last one minute of data was considered to determine the concentration for each standard by CRDS as each standard was flushed during 2 hours before.

P20, L464: *DAS* has to be defined.

P20, L474-478: It would be very interesting to know the origin of the problem and how the manufacturer solved it. As people from *Picarro* are co-authors of this paper it should be easy to have such information's.

P27, L631: *An excellent agreement was observed for measurements from both instruments…* Not so easy to check on Figure 18 (see the comment below).

P27, L634: …*contains large amount of water **and** $CO_2$ in addition to $O_2$*…

Figure 5, lower panel: Add the missing HDO lines (see HITRAN2016). For example the transition $4_{1\,3}$-$3_{1\,2}$ of the 111-000 band near 7878.500 cm$^{-1}$ has an intensity of $3.47\times10^{-27}$ cm/molecule.

Figure 14: What is the scale on the left?

Figure 18: The peaks corresponding to the flasks are not observed at the same time for the CRDS and IRMS. Moreover the delay between both experiments varies from one peak to the other. What is the reason for that? It will be better for comparison purposes to plot the two layers on the same graph.

---

## Referee Comment (RC2) · Anonymous Referee #2 · 26 Mar 2019

Review of manuscript "High-precision atmospheric oxygen measurement comparisons between a newly built CRDS analyzer (Picarro G-2207) and existing measurement techniques" by Tesfaye A. Berhanu et al. submitted to Atmospheric Measurement Techniques General comments: This paper from A. Tesfaye et al. presents the principle, the method and experimental tests conducted on a new CRDS analyzer dedicated to high precision oxygen measurements in the atmosphere and possibly additional measurement of isotopic content of O2. The first in-situ monitoring results obtained with this instrument are also presented and compared to other current existing measurements

technics running in parallel. In the introduction, the authors remind us about the scientific context and the scientific interest to measure O2 mixing ratio in the atmosphere, in the framework of carbon cycle budget and natural/anthropogenic source/sinks attributions, due to the strong link between the oxygen variability and the carbon cycle (combustion and respiration reactions). Then they highlight the analytical challenge to obtain a high precision measurement of O2 due to the very low level of atmospheric variability and they then shortly review the existing measuring technics currently available and the main experimental difficulties associated. In the second section, (Materials and methods), part 2.1, there is a description and discussion of the analyser design principles and characterizations (p4-14). The authors first describe the general instrument principle and design (including associated program modules), then explain the best conditions to be met for an ideal high precision measurement of molecular oxygen and finally constrains linked with an operational deployable field instrument. This provides them justification for the technical and methodological choices made such as spectroscopic model used, water vapour measurement and correction considerations, O2 measurement method design as well as O2 isotopic content measurement. On my opinion this section is a bit too long (about 1/3 of the full article) and also sometime a bit difficult to follow as a non-specialist of spectroscopy. I would suggest to shorten and simplify a bit this section if possible so that it can be more easy to follow. In the case it is not possible to shorten it I would recommend to modify the title of the article to better take into account this section which is anyway useful and interesting (but at the moment reading the title, I would expect the work to focus more on instrumental atmospheric data inter-comparison than technical and spectral analysis). The second subsection (part 2.2 and following) presents the instrumental tests and evaluation conducted in the laboratory at Picarro, at the University of Bern and in the field in Switzerland (two sites, Jungfraujoch and Beromünster). Experimental set up and conditions as well as methodologies adopted for the tests are presented in these subsections. The last section (section 3) presents and discuss the results of the different laboratory tests and in-situ monitoring. I would suggest to re-organize a bit this section with the previous

one. I think this would be easier for the reader to follow if the test results (in section 3) were merged together with the description of the tests procedure (in section 2). So I would merge 2.2.1 with 3.1.1, 3.2.2 with 3.1.2 (actually labelled 3.2 but should be 3.2.2) and also 2.3.3 and 3.2.3 (water correction). I would then keep all the in-situ parts together in section 3. In the last subsection, the authors present some results for test conducted with the analyser on the isotopic mode. The paper ends with a last concluding section. 2 One general comment and concern of this paper is the reporting unit used for O2 concentration all over the article. The authors used either the ppm unit (most of the time) or the per meg unit (also depending on the instrument used). As there are inter-comparison results used here to validate the new instrument but presented with a mix unit data, it is not easy to follow and to fully compare all data sets as well as precision of the different methods and instruments. Even though there is currently no official international unit to report O2/N2 mixing ratio, and also no Central Calibration Laboratory, there were recommendations given in the last WMO GAW report (report n°242) to report the O2/N2 mixing ratio in per meg units and also if possible to report it on the Scripps Institution of Oceanography (SIO). I would then suggest to make a choice of unit (preferentially per meg) all over the paper and present all the results in a uniform way. When necessary there is a relationship that might be used to express changes in O2/N2 ratio and equivalent changes in O2 mole fraction (Keeling et al, 1998; WMO GAW report n° 142). Having all number on the same unit would greatly help in the data comparison sections and table 1 (for example) except if this is not applicable. My general feeling about the paper is good, it is generally well written and most of the time clear. I would recommend this article for publication in AMT after revision, as this is a quite interesting new method to measure O2 with a great potential for atmospheric monitoring. Nevertheless, I would highly recommend to take into account the remarks and suggestions raised in the present review in order to straighten and improve the present manuscript. In particular, some additional calculation of mean values and standard deviations would help better evaluate the performances of the instrument compared to current ones. Specific comments: Abstract: Line 21: May need

to precise that the given short term precision given here refers to the O2 mixing ratio (not to the isotopic ratio). Line 21-23: In this sentence the authors state that the long term stability of the instrument is excellent and prevent high frequency calibration to assess an overall uncertainty of <5 ppm. The recommended calibration frequency is every 12h. With regards to my knowledge and own experience, paramagnetic technics only recall 24h calibration frequency to achieve similar precision. So I would suggest to moderate a bit this sentence, especially the beginning "In contrast to the currently existing techniques". The section two of the paper "material and methods" which is the longest part of the paper is not really mentioned in the abstract. May be a few more words should be added in the abstract to remind the reader about the work described in this later section. Introduction: The introduction section is not labelled as for the other sections (it should be section 1). Line 34-36: I would suggest to update the CO2 mixing ratio to the one of year 2017 (around 405 ppm). Line 47-51: There are also WMO/GAW precision recommendations and guidelines for O2/N2 ratio, as describe in the last GAW report (GAW report n°242, table 1 and p42-44). Line 55 and 56: Gas chromatography => gas chromatography Lien 57: As far as I know the techniques described in the previous sentence are not really commercially available. The sensors or detectors can be delivered by commercial companies but cannot be used directly to monitor O2 concentration. There is a need to "customize" these 3 detectors to build a monitoring instrument reaching the precision goal needed for atmospheric monitoring. This is most of the time done by the laboratories themselves! Line 61: I would add the following words at the end of the sentence: "... of the analysis method especially for continuous monitoring". Material and methods: ïĆů Analyser design principles: Line 79: Please define DFB Line 99-100: What is the typical range of variation for noise between the different instruments? Line 102: This is the first time that the Per meg unit it used in the paper (i.e. ppm is used most of the time). As already stated, it would be better to choose and harmonized the unit all over the paper. Line 116: I'm not an expert in spectroscopy, and the formalism used here to describe the absorption band is a bit unclear for me and a non-specialist. I don't know if there is a way of clarifying

or simplifying this another way? Line 143-145: This sentence is not clear, I suppose there is a verb missing: "In addition, the optical power in the ringdown cavity IS set by the ring-down detector threshold, which . . ." Line 159-160: This sentence is not clear: I think it should be "It stands out that the residuals that are largely an odd. . ." Line 185: Can the authors argue why they consider the dependence of Z on O2 too small to be significant? Line 196: please correct "for in measurements". Line 224: Can the author explicit what they mean by 161 isotopologue of water. This is absolutely not clear for a non-specialist in spectroscopy. Line 277: Please define "scm". Line 300: I would suggest to change ". . . adds additional .." to ". . . adds more. . ." Line 308-309: I'm a bit surprised that the instrument is providing a dry mole fraction for O2 using the water dilution experiment as it is stated by the authors above in the manuscript (line 273-274) "more works need to be done to investigate the water vapour correction of the oxygen measurement". My feeling is that the present day correction is still not fully satisfying and should be used with caution! There is also no direct explicit correction equation given in the text nor explanation on how this correction is implemented (or to they use the directly the linear function given on figure 7?). Line 327-330: Taking into account the low precision of the analyser for isotopic content as stated by the authors, is there still an interest to measure them within the context of environmental studies? Could the authors give us a few example and/or possible application of O2 isotope measurement in the environment that could be achieve with that instrument? ïĆů Laboratory tests at Picarro, Santa Clara: This section and following subsections should be relabelled, 2.2; 2.2.1; 2.2.2 etc. . . Line 346: Please define sccm. ïĆů Laboratory measurements at the University of Bern: Line 351-355: Could you please add a reference describing the Bern O2 analytic measurement systems (Both for The Fuel cell system and for the Mass Spectrometer) if available. Line 356: Could you please give us a bit more details about the "pressure controlling unit": What is it, What kind of flow meter? (short description or reference). 4 Line 367-377: I'm a bit surprised that there was no direct measurement made to the IRMS without the tee junction. To my knowledge the IRMS is the only one instrument that can provide a very high precision O2/N2 measurement and should be seen as the reference instrument. So I would have made the test in three steps, first with the Tee measuring on both instruments, then directly on the IRMS without the tee which would have given a reference value and then directly to the Picarro. All this at the different splitting ratios. Is there a reason why the direct measurement to the IRMS was not done? Line 380: please replace "case b" by "case ii". Line 378-389: The conclusion of this section are a bit disappointing as none of the results are shown and only one value is given (without uncertainty). Would it be possible to show the results of the tests? Previous studies by A. Manning have shown that the tee junction effect could be relevant at the level of precision that we are looking for atmospheric O2 monitoring. The impact given here (0.5ppm) is already more than half of the global precision stated for the instrument (<1 ppm line 23). So, if the instrument is to be commercialized, I would deeply recommend to go deeper into that question and firmly establish the conditions of use of a Tee junction or not. Line 403: How was established this correction function? What is the link with the test from figure 7? See also comment for line 308-309. Line 424: Add a reference to JFJ measurements and set up. Line 465: Please remove "is avoided" at the end of the sentence. Can the authors give us more precision about what they call "preconditioned"? Line 463-469: I have one question regarding this evaluation. Why do the authors use glass flasks? Why not connecting directly the CO2 free cylinder to the analytical device? This would avoid potential contamination during flask filling. Results and Discussions: See general comments for re-organisation proposition. Line 483-490: Looking at figure 9, it doesn't seem to me that there is a real drift. For me a drift would show a smooth continuous tendency to increase or decrease in the values. Here what I see is more something like a large variability on the measurement, I see an anti-correlation between O2 values et Y parameter and to a lesser degree a correlation between peak height and O2 as well. So I don't really understand what the authors mean by drift here. Could you please clarify. Are there also some ideas to eliminate or identify those small drift as stated on line 490? Line 511: What is a very good agreement? Can the authors give us an estimate of the mean difference (on a comparable unit for exam-

ple?). This would help to evaluate the accuracy of the new instrument and see how well it meets the WMO recommendation or not. Same goes for table 1 which is difficult to use because of the different units for the different instruments! Line 517-519: This is absolutely needed if the final goal is to get high precision O2 measurement. There is no need for high precision CO2 measurement but this dilution effect is to be taken into account as already done with present day "homemade" monitoring systems, even for atmospheric monitoring purpose. Line 521: Please change "The measurement precision of the Picarro G-2207 measurement was calculated..." to "The measurement precision of the Picarro G-2207 measurement was calculated..." Line 521-526: What about the precision of the reference instrument that should be the IRMS? Line 527-536: I fully agree that it is difficult to compare the graph as there is this problem of unit already highlighted in this review. I'm not sure how significant is the small difference in the correlation coefficient calculated here. 5 Line 541-542: I agree that the drift at the beginning could be linked with unstable condition after unpacking but the drift remains all over the measuring period and usually Picarro are stable within 3-5 hours after starting measurements. Line 551-553: Did the manufacturer find the cause of this drift. Was there any significant change in the hardware or software configuration of the initial instrument? Line 558-562: Did the authors also made water measurement comparisons between Licor and Picaro on wet air conditions? Line 564-565: This sentence is very confusing. Please reword as follow: "... in O2 measurements in both cases. (Figures 15c & 15d) shows in case..." Line 573-577: Can the authors provide some more precise numbers such as for example mean values and standard deviation calculated from data shown in figure 15c and 15d. This would greatly help quantify the variability and give a comparison element with regards to the given instrumental precision. Line 581-583: I disagree with this statement, there are several sections of the paper dealing with the water correction factor. There was a choice stated in the paper and made by the manufacturer to enable wet air measurement, so the water correction is a key issue if the instrument is going to be sold soon and to assess high precision measurement. I'm convinced the correction factor is not easy task to handle

and the results presented here are not sufficient to close the problem and give a final solution but this has to be further investigated. Line 585-593: How are the Picarro data calibrated (based on the in-situ calibration cylinders that have been measured also I suppose)? Could the author quantify a bit more precisely the "very good agreement" like for example providing the mean and the standard deviation of the data for both analysers over the full period. For me, based on the figure 16, it seems that there is a little offset between both instrument (paramagnetic a bit lower) and that there is a slight higher variability for the Picarro instrument compared to the other one but it is difficult to assess with only the figure. Line 595-607: I understand that the isotopic mode is not well suited for ambient O2 concentration measurement but what about the isotopic values? Any comment about those? Line 641: Can the authors provide a table with the individual values for each flasks and instrument so that we can really compare the results and evaluate the precision and repeatability of the measurements on each instrument? Can we add mean values and standard deviation for the three replicates? Line 644: I think the authors mean "...of water and CO2 in addition..." Line 653-655: Would the authors then recommend using the isotopic mode of the instrument at the moment (at least for atmospheric monitoring on atmospheric range) or still need some work to improve it and be sure it is reliable? (at least for atmospheric monitoring on atmospheric range)? Conclusion: Line 672-677: I feel that the conclusion driven here are a bit optimistic. It is stated several time in the paper that there is still work to do on this question. I would suggest to reword a little bit that conclusion in that way. Line 680-681: Here also I would like to see the data with mean values and standard deviation before drawing such an optimistic conclusion (see comments in the previous sections). I think this conclusion also lack a more general statement about the future applications of this instrument and possible improvement (especially for the isotopic mode). 6 Figure 1: The parameter $\tau$ is not define neither in the legend of the figure nor in the text. I wonder why the measurements presented here are made at 333Hpa and not at 340 hPa which is the nominal working pressure of the instrument (see line 86) ? Figure 9: There are strange value above each of the three upper graphs (+2.1028e5

on the upper one). What do they mean? Line 773: correct "shown to show".

---

## Author Comment (AC1) · 13 May 2019

**Reply to Reviewer 1 comments**

We would like to thank Reviewer 1 for his/her supportive and interesting comments. We provided here detailed explanations/comments/modification. For clarity, we kept the reviewer´s comments in red and our replies in Black colors.

General observations:

General comments

The paper entitled *High-precision atmospheric oxygen measurement comparisons between a newly built CRDS analyzer (Picarro G-2207) and existing measurement techniques* describes a Cavity Ring Down Spectrometer devoted to the determination of oxygen concentration in air and to the delta $^{18}O$ isotopic ratio measurement depending of the instrument mode. The performances of the instrument are tested in laboratory by comparing measurements of well-known samples with results obtained using other techniques (IRMS, paramagnetic technique, Licor instrument) on the same samples. In  the field measurements and comparisons are also provided by the authors at the High Altitude Research Station Jungfraujoch and at the Beromünster tall tower. The paper is well-written and detailed. The paper is now much easier to read thanks to the new way the experimental description and the obtained results are presented. The performances reached by the CRDS instrument are at  the state-of-the-art for optical methods. The paper is fully in the scope of AMTD and is well-suited for a publication in this journal but needs some significant corrections (see the comments below).

Main remarks:

Part of the work in Fleisher et al. 2015 to which the authors refer in lines 135-141 seems to not be published. If authors have a published reference corresponding to Fleisher et al. 2015 they can let  the text as it is but if they are not able to give a published reference they have to remove the lines 135-141 as well as the lines 144-146 and the reference.

The cited work by Fleisher is not yet a published article but exist as a conference paper. For this reason, we have now excluded this citation and its associated sections.

At the end of this paragraph we added the following sentence:

"Ultimately, the usefulness of the spectral model is to be evaluated by the precision and stability of the $O_2$ measurements when compared with established techniques."

The concentrations reported in Figures 8, 12, 13 have to be given in the same unit (per meg or better in ppm) to facilitate the comparison. For example in Figure 13 it will be better to plot the two layers (the paramagnetic and the CRDS measurements) on the same graph using the same units.

We used the units ppm and per meg to reflect the actual measured values by the specific analyzers as we believe it will be best to keep the reporting as closely connected as possible to what we directly measure, which is optical absorption. By reporting what we actually measure, our reported values also show most honestly whatever is missing from the picture, such as the dilution due to unmeasured

carbon dioxide. But for some sections, we have provided measurement values in the same unit for better comparison between different analyzers for example Figure 16 and Table 1.

Regarding the "specific comments":

P2, L33: The authors have to recall the definition of *per meg* unit.

We have now added this definition at lines 35-40 as follows:

Note that the variations in atmospheric O2 is expressed in units of per meg due to its small variations with respect to a large background and to account for dilution effects from CO2 or any other gas of relevant amount change, which is expressed as:

$$\delta\left(\frac{O_2}{N_2}\right)(per\ meg) = \left(\frac{(\frac{O_2}{N_2})_{sample}}{(\frac{O_2}{N_2})_{reference}} - 1\right).10^6 \qquad\qquad (1)$$

P2, L43: The authors have to recall the definition of *the oxidation ratio* and give some explanations.

Defined in lines 50-51 as:

"OR defined as the stoichiometric ratio of exchange during various process such as photosynthesis and respiration"

P4, L86: The authors have to specify the reference of the pressure gauge, the proportional valve and acquisition board allowing stabilizing the pressure at the level of 3×10$^{-5}$!
The pressure is stabilized such that the error signal of the pressure sensor is 3 x 10-5.  In other words, the actual pressure is (likely) not stabilized to that level, due to noise/drift in the sensor itself.

P4, L91: The authors should give the typical ring-down time they have.

We have now provided this information and added a sentence: "For this instrument the empty cavity ring-down time constant is about 39 μs."

P5, L104: Give here the FSR value instead of p7.

We provided the numerical value 0.0206 cm-1 in Line 105 and remove it from p7.

P6, L125: The authors should cite the following reference: Tran et al., JQSRT (2019) 222-223, p108- 114.

This reference is added Tran et al. 2019
Tran, H., M. Turbet, S. Hanoufa, X. Landsheere, P. Chelin, Q. Ma, and J.-M. Hartmann, 2019: The CO2-broadened H2O continuum in the 100-1500 cm-1 region. Measurements, predictions and empirical model. J. Quant. Spectrosc. Radiat. Transfer, 230, 75-80, doi:10.1016/j.jqsrt.2019.03.016.

P6, L142-144: This sentence is strangely written and should be reformulated.
As the reference to Fleisher and the discussion that goes with it are excluded, this part is also removed.

P7, L146-149: The excess noise observed by the authors is probably due to the fact that shorter ring-downs have less data points available for the fit. This should give a noise level increased by a factor  of τ^(-3/2) where τ corresponds to the RD time.

We have observed some excess noise on the ring-down time constants for the highest loss points at the peak of the Q13Q13 line that is greater than the expected tau^(-3/2) dependence, which might be caused by absorption that is not linear in optical power, but we cannot be certain of this explanation at present.

P7, L158: Put also on Figure 1 the residuals when a Voigt profile is used. Moreover, the residuals observed on this figure seem to be due to a frequency shift and not to the limitations of the Galatry model. In the description of the Galatry profile, the authors don't mention the collisional line-shift parameter. Is this parameter taken into account for the calibration procedure? What about line- mixing effect?

P7, L.158:  We have now revised Figure 1 considering the reviewer comment as follows:

[Figure]

Figure 1. The top panel (a) shows the raw data (points) and the best-fit Galatry function (solid line). Residuals of the Voigt fit are shown in panel (b) and residuals of the Galatry fit are shown in panel (c).

Regarding the additional questions: (1) Yes, we take the collisional line shift into account in our calibration procedure, and (2) it is entirely possible that line mixing affects the line shape, but we are not in the position to say with confidence to what extent it does. This belongs in the class of line shape phenomena that we consider to be outside the scope of our measurement and modeling abilities, but which we also do not think are essential for the operation of the analyzer.

P7, L166: *frequency of narrowing collisions* should be replaced by *velocity change collision rate*.
This sentence is now modified accordingly

P7, L167: The authors should be more precise: …$\sigma_D$ is the **1/e** Doppler **half**-width of the transition…

Corrected accordingly

P8, L169-170: The units of $k_B$, $T$, $M$ and $c$ have to be given.

The units of $k_B$, T, M and c are given as J.K$^{-1}$, K, amu and m/s, are now added to the manuscript

P8, L176: Cite HITRAN2016 instead of HITRAN2013.

We have now removed the citation of HITRA2013 and added HITRAN2016:

I.E. Gordon et al., The HITRAN2016 molecular spectroscopic database, Journal of Quantitative Spectroscopy & Radiative Transfer (2017), http://dx.doi.org/10.1016/j.jqsrt.2017.06.038 "

P8, L177: Give the uncertainty reported in HITRAN data base.
We have now added this information as "uncertainty code 4 for $\gamma_{air}$ corresponding to 10% --20% relative uncertainty".

P8, L178: Change the reference if required (see my comment above).

Updated to HITRAN 2016

P10, L217-219: Not true for the line near 7878.45 cm$^{-1}$ where the water transition dominates.

We would like to thank the reviewer for pointing out this important point. Indeed, we have not used the latest HITRAN2016 version water spectrum as this statement was derived from HITRAN2012. Hence:

- We added revised Figure 5, with a "stick plot" from Hitran2016 instead of Hitran2012.

[Figure]

Figure 5. Upper panel: spectrum of water in nitrogen (points) and fit to Voigt model (blue curve). Lower panel: Oxygen (green), normal water (blue), and deuterated water (red) lines in the 2016 Hitran data base."

- We also replaced the sentence that spans lines 217-219 as "The main features are the Q13Q13 line from trace contamination of oxygen in the sample and several lines that arise from normal water ($^1H_2^{16}O$, AFGL abbreviation 161) and deuterated water ($^1H^2H^{16}O$, AFGL abbreviation 162, also abbreviated HDO)."

P10, L219-220: Three lines of HDO are missing in the figure but are present in HITRAN2016.

This is now corrected in the new figure 5.

P10, L222-L224: This is not true as HDO lines are present.

This sentence is also now rewritten as: "The relative intensities of the 161 and 162 lines change with variations in the isotopic composition of the water, but fortunately the direct interference with the oxygen Q13Q13 lines comes entirely from the 161 isotopologue, with the strongest 162 line being separated by approximately 8 line widths (FWHM) from the Q13Q13 line."

P11, L245: *the z-parameter was constrained to be proportional to y, based on earlier measurements.* This is not clear for me. What are these earlier measurements and how they show that? Same remark for line 250.

We have now changed the words "earlier measurements" on line 245 to "measurements summarized in Figure 2". This figure shows that to a good approximation y and z are both proportional to pressure and therefore to each other when the pressure changes.

P11, L251-253: How the missing HDO lines will be treated as the HDO isotopic abundance is not determined from the 7817 cm$^{-1}$ window.

P11, L251-253:  The text describes what we did.  To address this we now added a sentence between the sentence that ends on line 253 and the sentence that begins on line 254 as follows:  "This procedure does not account for variations in HDO abundance, which may introduce some systematic error into the water vapor correction for samples of unusual isotopic composition, but it should accurately model the most important lines that interfere with the oxygen measurement."

P12, L277: *sccm* instead of *scm*.

Corrected accordingly

P13, L299-308: Maybe adding equations will make this paragraph easier to understand.
As there are in-line equations that clearly explain that by using the ratio of amplitude to line width rather than amplitude alone we obtain better precision in the determination of $O_2$ concentration, we do not see the importance of adding additional equations at this section.

P14, L326-329: What about the water lines in that spectral region (especially the $H_2O$ line near 7881.98 cm$^{-1}$)?

We are aware of these lines; they do not interfere strongly, as they are about two full-widths away from the line we measure.

P19, L436: It is strange that only the last one minute of data was considered to determine the concentration for each standard by CRDS as each standard was flushed during 2 hours before.
Selection of the last one-minute data done to be consistent with ambient air measurements which are usually switching from one height level to another and between standard cylinders usually within a couple of minutes.

P20, L464: *DAS* has to be defined.
We have now replaced the acronym DAS with "temperature inside the instrument chassis" as DAS seems to be too technical.

P20, L474-478: It would be very interesting to know the origin of the problem and how the manufacturer solved it. As people from *Picarro* are co-authors of this paper it should be easy to have such information's.
We have now included a possible hypothesis that could have led to such a drift to this line as follows:

"A possible hypothesis for the cause of the drift can be an optical amplifier in the first system and not anymore included in the design of the product which produced a significant amount of broadband light that could fill the cavity (albeit with a low coupling coefficient), and would ring down with a different (and generally much faster) time constant that the baseline loss of the cavity. However, the ringdown time on the peak of the oxygen line is just 10 microseconds, such that the broadband light might have distorted the single exponential decay of the central laser frequency, leading to the observed drift in the oxygen signal.  However, we were not able to confirm this hypothesis."

P27, L631: *An excellent agreement was observed for measurements from both instruments…* Not so easy to check on Figure 18 (see the comment below).

P27, L634: …*contains large amount of water **and** $CO_2$ in addition to $O_2$…*

Corrected as mentioned above

Figure 5, lower panel: Add the missing HDO lines (see HITRAN2016). For example the transition $4_{13} - 3_{12}$ of the 111-000 band near 7878.500 $cm^{-1}$ has an intensity of $3.47 \times 10^{-27}$ cm/molecule.
We have now modified Figure 5 as explained above

Figure 14: What is the scale on the left?
It is mmol/mol but uncalibrated values and now added to the Figure as shown below

[Figure]

Figure 14. Parallel water vapor measurements for a dried ambient air by both the NDIR and CRDS analyzers. Note that the water values from the NDIR analyzer are not calibrated.

Figure 18: The peaks corresponding to the flasks are not observed at the same time for the CRDS and IRMS. Moreover the delay between both experiments varies from one peak to the other. What is the reason for that? It will be better for comparison purposes to plot the two layers on the same graph.

This difference is simply due to the difference in time stamps from the two analyzers and the two are not plotted in the same figure as the purpose here is to provide a quantitative view of the peak signs. We have now added the time stamps for both plots in the x-axis (figure below).

[Figure]

Figure 18. Consecutive $\delta^{18}O$ measurements of a standard gas ($CO_2$-free air) filled into three flasks followed by measurement of breath air using the CRDS analyzer (top) and IRMS (bottom). These measurements were carried out in the middle of ambient air measurements.

---

## Author Comment (AC2) · 14 May 2019

Reply to Reviewer 2 comments

We would like to thank Reviewer 2 for his/her supportive and interesting comments. Unfortunately, these comments were made based on the first version of the manuscript, after we made thorough reorganization after suggestions by the Editor. It also makes it difficult understanding where these comments are located but we tried all our best. We provided here detailed explanations/comments/modification. For clarity, we kept the reviewer´s comments in red and our replies in Black colors.

Review of manuscript "High-precision atmospheric oxygen measurement comparisons between a newly built CRDS analyzer (Picarro G-2207) and existing measurement techniques" by Tesfaye A. Berhanu et al. submitted to Atmospheric Measurement Techniques General comments:

This paper from A. Tesfaye et al. presents the principle, the method and experimental tests conducted on a new CRDS analyzer dedicated to high precision oxygen measurements in the atmosphere and possibly additional measurement of isotopic content of O2. The first in-situ monitoring results obtained with this instrument are also presented and compared to other current existing measurements technics running in parallel. In the introduction, the authors remind us about the scientific context and the scientific interest to measure O2 mixing ratio in the atmosphere, in the framework of carbon cycle budget and natural/anthropogenic source/sinks attributions, due to the strong link between the oxygen variability and the carbon cycle (combustion and respiration reactions). Then they highlight the analytical challenge to obtain a high precision measurement of O2 due to the very low level of atmospheric variability and they then shortly review the existing measuring technics currently avail- able and the main experimental difficulties associated. In the second section, (Materials and methods), part 2.1, there is a description and discussion of the analyser design principles and characterizations (p4-14). The authors first describe the general instrument principle and design (including associated program modules), then explain the   best conditions to be met for an ideal high precision measurement of molecular oxygen and finally constrains linked with an operational deployable field instrument. This provides them justification for the technical and methodological choices made such as spectroscopic model used, water vapour measurement and correction considerations, O2 measurement method design as well as O2 isotopic content measurement. On my opinion this section is a bit too long (about 1/3 of the full article) and also sometime a   bit difficult to follow as a non-specialist of spectroscopy. I would suggest to shorten and simplify a bit this section if possible so that it can be more easy to follow.  In the case it   is not possible to shorten it I would recommend to modify the title of the article to better take into account this section which is anyway useful and interesting (but at the moment reading the title, I would expect the work to focus more on instrumental atmospheric   data inter-comparison than technical and spectral analysis).

We are aware that the spectroscopy section comprises significant part in this manuscript. However, we would like to keep these sections in the manuscript in line with Reviewer 1, who requested as much detail as possible about spectroscopy. We understood the need for reflecting this in the title of the manuscript but our main focus is still the intercomparison study between these analyzers.

The second subsection  (part 2.2 and following) presents the instrumental tests and evaluation conducted in the laboratory at Picarro, at the University of Bern and in the field in Switzerland (two sites, Jungfraujoch and Beromünster). Experimental set up and conditions as well as methodologies adopted for the tests are presented in these subsections. The last section (section 3) presents and

discuss the results of the different laboratory tests and in-situ monitoring. I would suggest to re-organize a bit this section with the previous one. I think this would be easier for the reader to follow if the test results (in section3) were merged together with the description of the tests procedure (in section 2).  So    I would merge 2.2.1 with 3.1.1, 3.2.2 with 3.1.2 (actually labelled 3.2 but should be   3.2.2) and also 2.3.3 and 3.2.3 (water correction).   I would then keep all the in-situ parts together in section 3.

We believe there is a small misunderstanding to which version of the manuscript these comments are provided. While we submitted our manuscript for the first time indeed we have these sections separated. But based on the recommendations of the Editor we have modified these sections similar to the comments given above. For example section 2.2.1 does not exist in the final version of the manuscript published here but rather merged to section 3.

In the last subsection, the authors present some results for test conducted with the analyser on the isotopic mode. The paper ends with a last concluding section. 2 One general comment and concern of this paper is the reporting unit used for O2 concentration all over the article.  The authors used either the ppm  unit (most of the time) or the per meg unit (also depending on the instrument used). As there are inter-comparison results used here to validate the new instrument but presented with a mix unit data, it is not easy to follow and to fully compare all data sets as well as precision of the different methods and instruments. Even though there is currently no official international unit to report O2/N2 mixing ratio, and also no Central Calibration Laboratory, there were recommendations given in the last WMO GAW report (report n[?] 242) to report the O2/N2 mixing ratio in per meg units and also if possible to report it on the Scripps Institution of Oceanography (SIO). I would then suggest to make a choice of unit (preferentially per meg) all over the paper and present all the  results in a uniform way. When necessary there is a relationship that might be used to express changes in O2/N2 ratio and equivalent changes in O2 mole fraction (Keeling et al, 1998;  WMO GAW  report n[?]  142).  Having all number on the same unit would greatly help in the data comparison sections and table 1 (for example) except if this is  not applicable.  My general feeling about the paper is good, it is generally well written  and most of the time clear. I would recommend this article for publication in AMT after revision, as this is a quite interesting new method to measure O2 with a great potential for atmospheric monitoring. Nevertheless, I would highly recommend to take into account the remarks and suggestions raised in the present review in order to straighten and improve the present manuscript. In particular, some additional calculation of mean values and standard deviations would help better evaluate the performances of the instrument compared to current ones.

We used the units ppm and per meg to reflect the actual measured values by the specific analyzers as we believe it will be best to keep the reporting as closely connected as possible to what we directly measure, which is optical absorption. By reporting what we actually measure, our reported values also show most honestly whatever is missing from the picture, such as the dilution due to unmeasured carbon dioxide.

In times where conversion is needed for comparison purpose, we provided the measurements from different analyzers in ppm unit for example Figures 15 &16.

Regarding Table 1, as it is clearly shown, we have provided all the oxygen values in per meg units and CO2 values in ppm.

Specific comments:

Abstract

Line 21: May need to precise that the given short term precision given here refers to the O2 mixing ratio (not to the isotopic ratio).

We have now specified this by modifying this sentence as "…standard error of one-minute $O_2$ mixing ratio measurements …."

Line 21-23: In this sentence the authors state that the long term stability of the instrument is excellent and prevent high frequency calibration to assess an overall uncertainty of <5 ppm. The recommended calibration frequency is every 12h. With regards to my knowledge and own experience, paramagnetic technics only recall 24h calibration frequency to achieve similar precision. So I would suggest to moderate a bit this sentence, especially the beginning "In contrast to the currently existing techniques".

This statement is partly correct. Indeed, a full calibration for the paramagnetic technique is also made every 18 hour but a frequent 18 minute offset correction is applied since the drift rate of a paramagnetic cell is immense despite a thorough control of the pressure and temperature. In our sentence we refer in particular to the short-term drift that is much better than for corresponding techniques and therefore, we would like to keep this sentence as it is.

The section two of the paper "material and methods" which is the longest part of the paper is not really mentioned in the abstract. May be a few more words should be added in the abstract to remind the reader about the work described in this later section.

We have now added a line to reflect this point in the abstract as follows:

"Here we present detailed description of the analyzer and its operating principles as well as comprehensive laboratory and field studies for a newly developed high-precision oxygen mixing ratio and isotopic composition analyzer (Picarro G-2207) that is based on cavity ring-down spectroscopy (CRDS)".

Introduction:

The introduction section is not labelled as for the other sections (it should be section 1).

It is now labelled as section 1

Line 34-36: I would suggest to update the CO2 mixing ratio to the one of year 2017 (around 405 ppm).

Modified to 405.0 ppm

Line 47-51: There are also WMO/GAW precision recommendations and guidelines for O2/N2 ratio, as describe in the last GAW report (GAW report n ∘ 242, table 1 and p42-44).

We have now added this information on Line 70 as:

Note that the GAW recommendation for the measurement precision of $O_2/N_2$ is 2 per meg.

Line 55 and 56: Gas chromatography => gas chromatography

Corrected as suggested

Lien 57: As far as I know the techniques described in the previous sentence are not really commercially available. The sensors or detectors can be delivered by commercial companies but cannot be used directly to monitor O2 concentration. There is a need to "customize" these 3 detectors to build a monitoring instrument reaching the precision goal needed for atmospheric monitoring. This is most of the time done by the laboratories themselves!

We agree with the reviewer here. Instruments capable of making O2 measurements of the well-mixed atmosphere at anything close to the precision needed for the scientific goals of the atmospheric community are neither commercially available nor widely used. These are custom-built analyzers that require a great deal of expertise to set up and run them, and to interpret the results properly.

We have removed part of the sentence "commercially available" and the beginning of this paragraph now reads as:

"Currently there are several techniques mostly custom built that can measure…."

Line 61: I would add the following words at the end of the sentence: "... of the analysis method especially for continuous monitoring".

We have now added "…especially for continuous monitoring" to this section.

Material and methods:

Analyser design principles:

Line 79: Please define DFB

DFB signifies Distributed Feedback and this term used now in the manuscript instead of DFB.

Line 99-100: What is the typical range of variation for noise between the different instruments?

As far as noise-equivalent absorption goes, that varies by as much as a factor of two between instruments.

Line 102: This is the first time that the Per meg unit    it used in the paper (i.e. ppm is used most of the time). As already stated, it would be better to choose and harmonized the unit all over the paper.

We have now defined the per meg unit with equation as follows in the manuscript:

Note that the variations in atmospheric O2 is expressed in units of per meg due to its small variations with respect to a large background and to account for dilution effects from CO2 or any other gas of relevant amount change, which is expressed as:

$$\delta\left(\frac{O_2}{N_2}\right)(per\ meg) = \left(\frac{(\frac{O_2}{N_2})_{sample}}{(\frac{O_2}{N_2})_{reference}} - 1\right).10^6 \qquad\qquad (1)$$

Line 116: I'm not an expert in spectroscopy, and the formalism used here to describe the absorption band is a bit unclear for me and a non-specialist. I don't know if there is a way of clarifying or simplifying this another way?

We provided the details of transitions we measure with the quantum numbers of the states measured. However, the main concept here, for a non-spectroscopist, is that we measure a single, isolated absorption line in the 1.27 micron band.

Line 143-145:  This sentence is not clear, I suppose  there is a verb missing: "In addition, the optical power in the ringdown cavity IS set by  the ring-down detector threshold, which . . ."

This section is now excluded from the manuscript as suggested by Reviewer 1

Line 159-160:  This sentence is not clear:     I think it should be "It stands out that the residuals that are largely an odd. . ."

We clarified this sentence as:

"It stands out that the residuals are largely odd in detuning from the line center…"

Line 185: Can the authors argue why they consider the dependence of Z on O2 too small to be significant?

The measurements show that any variation is at most comparable to the error bars, so we do not consider it a significant effect.

Line 196: please correct "for in measurements".

The word "in" is now removed

Line 224: Can the author explicit what they mean by  161 isotopologue of water.  This is absolutely not clear for a non-specialist in spectroscopy.

This sentence is now modified as follows:

The main features are the Q13Q13 line from trace contamination of oxygen in the sample and several lines that arise from normal water ($^1H_2^{16}O$, AFGL abbreviation 161) and deuterated water ($^1H^2H^{16}O$, AFGL abbreviation 162, also abbreviated HDO).

Line 277: Please define "scm".

It is now corrected as "sccm" meaning Standard Cubic Centimeter per Minute

Line 300: I would  suggest to change ". . . adds additional .."   to ". . . adds more. . ."

Now modified accordingly

Line 308-309:  I'm a   bit surprised that the instrument is providing a dry mole fraction for O2 using the water dilution experiment as it is stated by the authors above in the manuscript (line 273-274) "more works need to be done to investigate the water vapour correction of the oxygen measurement".  My feeling is that the present day correction is still not fully satisfying and should be used with caution! There is also no direct explicit correction equation  given in the text nor explanation on how this correction is implemented (or to they use  the directly the linear function given on figure 7?).

Complete validation of the water vapor correction has not yet been performed, and is beyond the scope of this paper.

As for the numerical details, the linear fit in Figure 7 determines a linear relationship between optical absorption and water fraction, and the correction to oxygen is just the usual dilution correction.

Line 327-330: Taking into account the low precision of the analyser for isotopic content as stated by the authors, is there still an interest to measure them within the context of environmental studies? Could the authors give us a few example and/or possible application of O2 isotope measurement in the environment that could be achieve with that instrument?

We are not quite sure whether we understood the reviewer's comment correctly. Therefore, we refer to the two issues we can think of. First, regarding the degraded precision of the concentration measurement in the isotope mode. This is due to choosing a weaker main oxygen line to be closer to the minor isotope line selected and a significantly reduced number of ring-downs for the main oxygen line to favor the precision of the minor isotope line. Here, a further optimization depending on the users' needs is possible. Secondly, the interference on the isotope ratio itself on breath air is not yet understood. Further measurements are required to see which substance or substances are responsible for this interference. We would like to mention, though, that measurements on the compressed air composition led to a good agreement. Therefore, we can think of the following applications in the field of environmental research. Biological applications relevant for the climate and environmental research, i.e. photosynthesis/respiration processes close to the plants or even using leaf chambers. Analysis of vertical profile air samples taken by means of an AirCore is another application. First measurements have been taken in 2018. Here, stratospheric-tropospheric differences can be a focus. Many other process studies can be thought of where the oxygen is involved, e.g. combustion processes, electrolysis where incompleteness of the process will lead to isotope anomalies.

Another important application is in isotopic tracer experiments, in which either isotopically labeled carbon dioxide or water can be introduced into a closed plant system to understand better the photosynthesis. The isotope labeling can be performed at levels where the signals are greater than the errors in the measurement.

Laboratory tests at Picarro, Santa Clara:

This section and following subsections should be relabelled, 2.2; 2.2.1; 2.2.2 etc. . .

These sections have been modified and merged to section 3 of the manuscript as suggested by the Editor

Line 346: Please define sccm.

Defined above

Laboratory measurements at the University of Bern:

Line 351-355: Could you please add a reference describing the Bern O2 analytic measurement systems (Both for The Fuel cell system and for the Mass Spectrometer) if available.

We have now added the following reference:

Sturm, P., M. Leuenberger, F.L. Valentino, B. Lehmann, and B. Ihly, Measurements of $CO_2$, its stable isotopes, $O_2/N_2$, and [222]Rn at Bern, Switzerland, *Atmospheric Chemistry and Physics*, *6*, 1991-2004, 2006.

Line 356:  Could you please give us a bit more details about the "pressure controlling unit": What is it, What kind of flow meter? (short description or reference).

The pressure control system includes an electronic controller (Type 250E, MKS) which maintains a pressure difference of zero (precise to 0.005 mbar) across the pressure transducer (Baratron 223, MKS) by adjusting the waste flow using the nearby solenoid control valve (Type 248, MKS)

Line 367-377:  I'm a bit surprised that there was no direct measurement made to the IRMS without the tee junction. To my knowledge the IRMS is the only one instrument that can provide a very high precision O2/N2 measurement and should be seen as the reference instrument. So I would have made the test in three steps, first with the Tee  measuring on both instruments, then directly on the IRMS without the tee which would have given a reference value and then directly to the Picarro. All this at the different splitting ratios. Is there a reason why the direct measurement to the IRMS was not done?

Actually, there is no specific reason why a direct measurement was not conducted on the IRMS. However, as we conducted comparison of the CRDS analyzer and the IRMS by directly measuring multiple standard gases (See Table 1), we believe it can provide an excellent estimate of how the CRDS measurements are comparable to the IRMS.

Line 380: please replace "case b" by "case  ii".

Corrected to Case ii

Line 378-389:  The conclusion of this section are a bit disappointing as none of the results are shown and only one value is given (without uncertainty). Would it be possible to show the results of the tests? Previous studies by A. Manning have shown that the tee junction effect could be relevant at the level of precision that we are looking for atmospheric O2 monitoring.  The impact given here (0.5ppm) is already more  than half of the global precision stated for the instrument (<1 ppm line 23). So, if the instrument is to be commercialized, I would deeply recommend to go deeper into that question and firmly establish the conditions of use of a Tee junction or not.

We agree with the reviewer that it is important to further investigate the tee junction influence on the O2 measurements. During this test period, we have tried to test different scenarios of splitting ratios effect. Unfortunately, our observations are inconclusive which is mostly attributed to the temperature effect observed while decanting a cylinder at high flow. Note that the CRDS analyzer takes about 45ml/min and if we would like to go for a splitting ratio of 1:100, we need to decant the cylinder at a flow of 4.5 L/min, which led to cooling effect at the cylinder gauge. As A. Manning has also shown that temperature plays a major role in fractionating oxygen.  Meanwhile, the analyser is commercially available and we ask users to make their own tests or use split ratios if needed in the range where we document the values in the manuscript.

Line 403: How was established this correction function? What is the link with the test from figure 7? See also comment for line 308-309.

Already replied above

Line 424: Add a reference to JFJ measurements and set up.

We have now added the requested reference:

Schibig, M. F., Steinbacher, M., Buchmann, B., van der Laan-Luijkx, I. T., van der Laan, S., Ranjan, S. and Leuenberger, M. C,: Comparison of continuous in situ $CO_2$ observations at Jungfraujoch using two different measurement techniques, Atmospheric Measurement Techniques , 8, 57-68, 10.5194/amt-8-57-2015, 2015.

Line 465: Please remove "is avoided" at the end of the sentence. Can the authors give us more precision about what they call "preconditioned"?

The word "is avoided" is now removed

Preconditioning is a standard procedure at our lab for all flask samples prior to using them for sampling. A dedicated vacuum line was used to pump these flasks to vacuum, then flush multiple times with dry air and fill them to 1 atm with this air prior to sampling.

Line 463-469: I have one question regarding this evaluation. Why do the authors use glass flasks? Why not connecting directly the $CO_2$ free cylinder to the analytical device? This would avoid potential contamination during flask filling.

Its simply because it is easier to for experimental set up for example as we were placing these flasks before and after a water trap, which cannot be easily done with the cylinder.

Results and Discussions: See general comments for re-organisation proposition.

As we mentioned in the sections above, these comments were made on the first version of the MS and these reorganizations have already been applied.

Line 483-490: Looking at figure 9, it doesn't seem to me that there is a real drift. For me a drift would show a smooth continuous tendency to increase or decrease in the values. Here what I see is more something like a large variability on the measurement, I see an anti-correlation between O2 values et Y parameter and to a lesser degree a correlation between peak height and O2 as well. So I don't really understand what the authors mean by drift here. Could you please clarify. Are there also some ideas to eliminate or identify those small drift as stated on line 490?

It should be noted here that the measurement that the instrument reports is not Gaussian in nature, such that the Allan std. deviation does not decrease according to the square root of averaging time for long times (> 1 hour).

To clarify more this paragraph and in agreement with the reviewer's comment, we have now replaced the word 'drift' with 'error' as "the residual error of the analyzer…." And "Possible sources of error……".

Line 511: What is a very good agreement? Can the authors give us an estimate of the mean difference (on a comparable unit for example?). This would help to evaluate the accuracy of the

new instrument and see how well it meets the WMO recommendation or not. Same goes for table 1 which is difficult   to use because of the different units for the different instruments!

We intentionally converted the O2-CRDS values from ppm to per meg units to make them comparable. As can be seen from table 1 the three Scripps cylinders (ST3-ST5) IRMS UBern measurements are in agreement with the assigned values by Scripps to 0.6 ± 3.7 per meg, the O2-CRDS measurements for those 5.6 ± 17.8 per meg and the O2-Parameagnetic measurements show a comparison of 20.6 ± 26.8 per meg. A similar agreement is obtained between the three methods when including the cylinders ST-1 and ST-2 prepared by UBern. The picture is different for the ST-6 and ST-7 for known reasons as explained in the manuscript.

Line 517-519:  This    is absolutely needed if the final goal is to get high precision O2 measurement. There    is no need for high precision CO2 measurement but this dilution effect is to be taken   into account as already done with present day "homemade" monitoring systems, even  for atmospheric monitoring purpose.

Here we are referring to a cylinder with 2700 ppm CO2! As we mentioned in multiple sentences including the conclusion section, it will be important to have a parallel CO2 measurement (or the possibility to have a second laser for CO2, at least in the future) to account for dilution effect. As a side note any kind of gas addition to an ambient air will lead to a gas dilution effect. This even includes using compressed air by gas filling company or self-made ambient air compression when there small leak in the compression line, which could alter the gas mixture. Which could, for instance, lead a change in Ar/N2 or O2/N2. Even more care should be taken when using artificially compressed air-like gas mixtures. Here a proper determination of the gas components needs to be done.

Line 521: Please change "The measurement precision of the Picarro G-2207 measurement was calculated. . ." to "The measurement precision of the Picarro G-2207 measurement was calculated. ."

This comment is not clear. But we changed the text as:

"The measurement precision of the CRDS analyzer was calculated. . ."

Line 521-526: What about the precision of the reference instrument that should be the IRMS?

See answer to the question above (line 511…)

Line 527-    536: I fully agree that it is difficult to compare the graph as there is this problem of unit already highlighted in this review.  I'm not sure how significant is the small difference in the correlation coefficient calculated here.

The fact that the mean offset as well as the standard deviation of the measurements of ST-1 to ST5, as given above in the answer to question line 511…., is larger for the paramagnetic cell (at least for the instrument at UBern) than for the CRDS instrument, can easily clarify our statement in this line.

Line 541-542:  I agree that the drift at   the beginning could be linked with unstable condition after unpacking but the drift remains all over the measuring period and usually Picarro are stable within 3-5 hours  after starting measurements. Line 551-553:  Did the manufacturer find the cause of   this drift. Was there any significant change in the hardware or software configuration of the initial instrument?

One possibility for the cause in the drift was an optical amplifier in the system, which produced a significant amount of broadband light. This light could fill the cavity (albeit with a low coupling coefficient), and would ring down with a different (and generally much faster) time constant that the baseline loss of the cavity. However, the ringdown time on the peak of the oxygen line is just 10 microseconds, such that the broadband light might have distorted the single exponential decay of the central laser frequency, leading to the observed drift in the oxygen signal. We were however not able to confirm this hypothesis. There is no optical amplifier in the present design of the product.

Line 558-562: Did the authors also made water measurement comparisons between Licor and Picaro on wet air conditions?

We have made all the LICOR measurements using wet samples. However, we did not make an absolute comparison between the LICOR and CRDS analyzers for two reasons:

- First the LICOR water measurements are not calibrated (it of course could be done but firstly this would be outside of the scope of this publication and secondly we generally dry the ambient air)

- Second our focus in this manuscript was not comparing the water measurements by both devices

But as it can be seen in Figure 14, the water measurements from both analyzers for dried and non-dried air show similar behavior with matching water peaks

Line 564-565: This sentence is very confusing. Please reword as follow: ". . . in O2 measurements in both cases. (Figures 15c & 15d) shows in case. . ."

This section is now rephrased as follows:

"The water correction test was conducted by measuring dried ambient air (Figure 15a) into both analyzers as well as allowing non-dried air to the CRDS analyzer only (Figure 15b) and comparing the difference in $O_2$ measurements in both cases. Figures 15c & 15d show the water contents of dried ambient air measured in both analyzers (note that the CRDS uses its in-built water correction function)."

Line 573-577: Can the authors provide some more precise numbers such as for example mean values and standard deviation calculated from data shown in figure 15c and 15d. This would greatly help quantify the variability and give a comparison element with regards to the given instrumental precision.

For 15 c, mean = 1.85, Standard deviation=6.8

For 15 d, mean = -10.4, Standard deviation=14.6

Line 581-583: I disagree with this statement, there are several sections of the paper dealing with the water correction factor. There was a choice stated in the paper and made by the manufacturer to enable wet air measurement, so the water correction is a key issue if the instrument is going to be sold soon and to assess high precision measurement. I'm convinced the correction factor is not easy task to handle and the results presented here are not sufficient to close the problem and give a final solution but this has to be further investigated.

We fully agree with the reviewer that further and even more detailed and extended water correction analyses have to be performed, but we do not agree that it should be part of this publication. We note that for the most of the instruments on the market further improvement of correction functions are found over time. This will certainly also be the case here.

Line 585-593: How are the Picarro data calibrated (based on the in-situ calibration cylinders that have been measured also I suppose)? Could the author quantify a bit more precisely the "very good agreement" like for example providing the mean and the standard deviation of the data for both analysers over the full period. For me, based on the figure 16, it seems that there is a little offset between both instrument (paramagnetic a bit lower) and that there is a slight higher variability for the Picarro instrument compared to the other one but it is difficult to assess with only the figure.

Yes, the Picarro data is calibrated using the standard cylinders measured in-situ.

This question is a bit unclear. These are ambient air measurements with natural variability (on top of the variability from the analyzers) and we do not see the point of providing the mean and sd of these measurements.

If what the reviewer is implying here is for the difference, the calculated mean is -0.33 ppm and a standard error of 0.11 ppm.

Line 595-607: I understand that the isotopic mode is not well suited for ambient O2 concentration measurement but what about the isotopic values? Any comment about those?

Yes, this in an interesting question. The isotope values of ambient air after calibration using internal standards corresponds to expectations. The short-term (second) variations are large but the standard deviations of 5 minute means corresponds to about 0.3 permil.

Line 641: Can the authors provide a table with the individual values for each flasks and instrument so that we can really compare the results and evaluate the precision and repeatability of the measurements on each instrument? Can we add mean values and standard deviation for the three replicates?

We do not think it is relevant to include these values in the manuscript as we already stated that the isotope measurements from the CRDS analyzer needs a closer look but the plots in Figure 18 already gives a clear idea about the above mentioned topics.

Line 644: I think the authors mean "...of water and CO2 in addition..."

Corrected accordingly

Line 653-655: Would the authors then recommend using the isotopic mode of the instrument at the moment (at least for atmospheric monitoring on atmospheric range) or still need some work to improve it and be sure it is reliable? (at least for atmospheric monitoring on atmospheric range)?

This analyzer cannot measure natural isotopic variations. But, it can be used in tracer experiments where artificially enriched isotopes are used to study various biological processes such as photosynthesis. However, the manufacturer will continue to work on improving it.

Conclusion:

Line 672-677: I feel that the conclusion driven here are a bit optimistic. It is stated several time in the paper that there is still work to do on this question. I would suggest to reword a little bit that conclusion in that way.

We have now added the following sentence recommending for further tests about the water correction.

"However, we believe that it is important to further investigate this issue and identify an improved water correction strategy."

Line 680-681: Here also I would like to see the data with mean values and standard deviation before drawing such an optimistic conclusion (see comments in the previous sections). I think this conclusion also lack a more general statement about the future applications of this instrument and possible improvement (especially for the isotopic mode).

We have now added the following sentence at the end of this paragraph:

"However, we believe that this analyzer can be used for tracer experiments where artificially enriched isotopes are used to study biological processes such as photosynthesis in plants using isotopically labelled $CO_2$ and $H_2O$".

Figure 1: The parameter $\tau$ is not define neither in the legend of the figure nor in the text. I wonder why the measurements presented here are made at 333Hpa and not at 340 hPa which is the nominal working pressure of the instrument (see line 86)

Tau is just the averaging time for the Allan variance. All measurements with this analyzer were made at 255 Torr, which are same as 340 hPa.

Figure 9: There are strange value above each of the three upper graphs (+2.1028e5 on the upper one). What do they mean?

This is a notation to indicate an offset of the y-axis. In other words, the oxygen fraction reported in the top panel of what is now Figure 8 varies by about 12 ppm about a value of 2.1028e-5.

Line 773: correct "shown to show".

Corrected as "show"

---

## Author Response (AR2)

Replies to the Associate Editor (Christof Janssen)

We would like to thank the editor for his thorough comments/suggestions. Below are the replies to his questions and the modifications we added to the manuscript. The editor's comments are highlighted in red font while our replies are in black font.

Comments to the Author:
Dear authors, I congratulate you to your very interesting work that the referees have attested to be scientifically sound. Thank you also for responding to their requests in detail. I have thoroughly reread the revised manuscript and there is still a (long) list of minor corrections that should be addressed before publication (see further below). There are also four important issues that should be adressed in a further revision process:

P10 L227-228 In the case of CO2, the dilution of oxygen due to 400 ppm of CO2 is significant, and larger than any direct spectral interference.

I don't think you give detailed information (in form of an equation) on how the analyzer obtains its result from the measured quantities. The CRDS methd (and you could recall the fundamental equation) provides direct access to the particle number density of O2 (knowing the spectral data). How is the mixing ratio X_O2 obtained ? In principle you could use p and T, but I think this is not precise enough and some calibration is employed. Please give an equation or a reference where one sees how measured quantities (after correction or not) give O2 mixing ratios. (This would also allow to understand where the dilution effect comes from). The same holds for the conversion into O2/N2: what is the assumption on N2 (or f_N2) made in your algorithm to convert from X_O2 to O2/N2. What are the roles of water and CO2 ? Your data show that this is crucial information that is yet missing. Since your paper is the first description of such a system, the information must be provided.

The relationship of the measured quantities to the reported result is described very explicitly in lines 308—312 of the MS. As the reviewer mentions, the number density is directly proportional to the optical absorption, and for controlled temperature and pressure the number density is proportional to the mole fraction.  It is correct that we cannot measure temperature and pressure to the permeg level, hence calibration is required and this is actually the heart of the paper. We know that there are dilution effects from CO2 and water.  Water can be removed by drying but since we have no way to measure CO2, any dilution correction has to come from an independent measurement.  Unlike mass spectrometry, we do not measure O2/N2 directly; it can only be inferred by calibrating with standards. None of this should be remarkable – the same considerations apply to any optical method, whether it is diamagnetism or refractive index or absorption or whatever.  The key is a quantitative measure of oxygen concentration that is precise, stable, and calibratable at the permeg level.

Based on the evidence that you provide in this paragraph you should consider writing "The most likely cause for the drift is the optical amplifier ..." or "We believe that the optical amplifier has caused the drift ... instead of "A possible hypothesis for the cause of the drift can be an optical emplifier ..."

In line 504, we have now modified the sentence as:

"We believe that the optical amplifier has caused the drift in the first system…"

The section describes tests of the systems' water correction function.
You should consider organisation/presentation of this paragraph. If I strongly summarize and exaggerate this paragraph, then the water correction works if there is no water and it does not if there is water in the samples.

This section is now reorganized and modified mainly in the paragraph below:

Figure 15a shows the dried ambient air water measurements in both analyzers with frequent spikes due to valve switching while measuring standard gases. In the second case, where the water trap was by-passed and non-dried air was allowed to the CRDS analyzer keeping the dried air flow to the NDIR (Figure 15b), a clear increase in the water measurements in the CRDS analyzer can be observed. Here, it should be noted that there are no spikes in the water measurements of the CRDS analyzers as there are no standard gas measurements in between and the inlet is directly connected to the CRDS analyzer (Figure 11).  Figures 15c & 15d show the difference in oxygen measurements  of ambient air measured in both analyzers in the two cases stated above (note that the CRDS uses its built-in water correction function applying Eq. 5).

Figures/units
Please use a consistent notation to indicate units in axis labels. Most of the time units are given in parantheses, but some figures use brackets (Figs. 2, 5, 8, 9, 10), some use a slash (Fig. 7), some don't use anything at all (Fig. 17), and some even mix brackets and parantheses (Fig. 1).

These comments are now addressed as pointed out here with all the units in Parenthesis

List of detailed suggestions/corrections

P1 L16-17: Change to : Here we present a detailed description ...

The missing "a" is now added

P2 L35-40: Please check grammar, also consider that the choice of the unit (per meg) and the choice of the measurement quantity (O2/N2) are two different things. It would be better first to introduce the quantity O2/N2 and explain why and then the corresponding unit (small relative deviation from a standard).

As the editor´s suggestion we have now added the following paragraph and modify this section:

Measurements of atmospheric O2 are reported as the ratio to the N2 concentration and expressed as δ(O2/N2) because the variations in the concentrations of other atmospheric gases such as CO2 can influence the O2 partial pressure while this ratio is insensitive to these changes in other gases. These variations in atmospheric O2 is commonly expressed in units of per meg due to its small variations with respect to a large background, where

$$\delta\left(\frac{O_2}{N_2}\right)(per\ meg) = \left(\frac{(\frac{O_2}{N_2})_{sample}}{(\frac{O_2}{N_2})_{reference}} - 1\right) \cdot 10^6$$

P2 L40+ Please give a conversion formula that converts between the different units per meg and ppm (could be in the appendix). This would enable to better understand your results (eg. correlation plots in appendix).

We convert per meg to parts per million equivalent by multiplying per meg by 0.209500 (the O2 mole fraction of atmospheric air)

P3 L49-53: One gets confused about alphas and ORs in this paragraph. Please define. Use comma after OR. What is alpha ? Has it been defined before ?

This section is now rephrased to include OR and alpha definition as:

This method hinges on the linear coupling between CO2 and O2 with an oxidation ratio (OR, defined as the stoichiometric ratio of exchange during various process such as photosynthesis and respiration expressed using α) of 1.1 for the terrestrial biosphere photosynthesis-respiration processes (αb) and 1.4 for fossil fuel combustion (αf) while they are decoupled for oceanic processes (αo = 0).

P3 L52: give alpha_ocean for the decoupled process (alpha_o = 0)

See above

P3 L59: ... several, mostly custom built techniques ...

Corrected accordingly as ... several, mostly custom built techniques ...

P3 L69: monitoring(Note -> monitoring (note

Small n is now used

P4 L87: If you capitalize for mentioning the acronym of a DFB laser, please also capitalize the B in feedback

B is now capitalized

P5 L96-98: A drawing of the optical setup is missing. Please state briefly how the wavelength monitor is integrated in the optics (beam splitter, ...).

The following paragraph is now added:

"The wavelength monitor is a fiber-coupled device located between the laser and the cavity. A fraction of the beam from the input fiber is collected using a beam splitter for the measurement wavelength and the remaining power is collected in the output fiber."

P5 L98: It is a little bit strange to read : "the data acquisition system sweeps the laser frequency ...".

This sentence is now modified as:

"The instrument´s data acquisition system is used to sweep the laser frequency…."

P6 L100 (P9 L 198, 202): Strictly speaking, 7878.805547 cm-1 is no frequency, remove "a frequency of"

Removed "…a frequency of…" at these lines

P10 L221-224: It is not clear what you want say here. The H2O line clearly needs to be considered even if its line strength is only 10 to 20 % of the O2 line.

We do consider interference from water, as the text clearly states.  The point of ll. 221-224 is that we do not have to worry about deuterated water interfering with the O2 measurement because these lines are well separated.

P10 L217-228: There seems to be little advantage in using AFGL notation. General understanding would greatly benefit from dropping this notation and call isotopologues by their names: water and deuterated water or by their isotope formula ^1H_2^16O etc.

The AFGL notation is useful because of its conciseness and for clarity we have provided a clear definition of these on lines 237-239 as:

"The main features are the Q13Q13 line from trace contamination of oxygen in the sample and several lines that arise from normal water ($^1H_2^{16}O$, AFGL code 161) and deuterated water ($^1H^2H^{16}O$, AFGL code 162, also abbreviated HDO)."

P10 L218: I could not find traces of the heavy water isotopologues in the lower panel of Fig. 5. It seems that the figure has not been updated in the revised version, contrary to what has been indicated in the response to the reviewers. The caption also lacks the update.

The correct figure is now provided as shown below.

[Figure]

Figure 5. Upper panel: spectrum of water in nitrogen (points) and fit to Voigt model (blue curve). Lower panel: Oxygen (green), normal water (blue), and deuterated water (red) lines in the 2016 Hitran data base.

P10 L236: AFGL abbreviation -> AFGL code; please see remarks in the beginning.

See above

P11 L255-259: The description is misleading, because line profiles (area = 1) are not multiplied by amplitude, but by the line strength. Please phrase differently. Also what are the weak perturbing peaks ?

This comment is a bit unclear this sentence states exactly the procedure that was used. Our line profiles (which are not defined the same way as Hitran's, but are very explicitly explained in ll. 236-238) and our amplitudes are self-consistent. We specifically use the word "amplitude" and not "line strength" because we are describing our fitting procedure and not a Hitran calculation.

However, we modified the section "three water peaks and the two weak perturbing peaks" as

"the water spectrum is modeled with three peaks: one strong line and two weak perturbers", as described in lines 198-202 on p. 9.

P11 L260: I don't understand the phrase "that constrained to be in fixed ratios to be in fixed ratios". Is there a "were" missing ? "ratio" should probably be replaced by "proportion".

Corrected by adding "were constrained" and "ratio" is replaced by "proportion" (lines 263-264)

P12 L259-260: You might look up the recent HITRAN update.

This comment is a bit unclear. We used Gordon et al. 2017 for referring to the HITRAN in this MS and we will be glad to know if there is any update that is important and we are not aware to include in this section.

P12 L263: Please use here and at other instances the notation Filges et al. (2018) and not Filges et al. (Filges et al., 2018) when citing the author in the text.

Now corrected as Filges et al. (2018)

P12 L252: The water lines … change , into ; before rather.

Changed to "rather;…"

P15 L322-324: check passive voice in "there is no advantage to be obtained" and the whole phrase. It does not sound correct.

In line 343, we have now modified this phrase as:

"One is that in determining an isotopic ratio, normalizing absorption amplitudes to line widths does not provide any advantage, instead we simply take the ratio of amplitudes to compute delta."

P15 L346: Giving the absolute value of 5e-9 cm-1 does not help in the argument. 1. there are CRDS systems that measure lower absorptions. 2. You don't give an equivalent value for an absorption that is easily measurable (such as the 16O16O isotopologue).

Our statement is valid as stands – signal-to-noise is not adequate to measure amplitude and width independently.

As to point 2, the absorption of the 16O16O isotopologue is clearly shown in Figure 1.

P15 L333: Is much less precise: It should be sqrt(305/40) ~ 3 times less precise. Is this much less precise ?

The line intensity for the 16O16O measurement is also less for the isotopic mode than for the concentration mode, which makes the precision even worse than the factor of 3 computed above. The isotopic mode was not intended to give precise mole fraction.

P15 L337-339: The phrase "One set of tests aimed at determining the impact of pressure or temperature drifts or of uncobtrolled noise on the concentration measurements" ? can be much simplified to ease reading. What is uncontrolled noise ? This needs to be clarified.

We have now removed the word "uncontrolled" in line 362.
P15 L353: Drift -> drift

Corrected as drift
P15 L356: insolated ?

It is now corrected as "insulated"
P15 L358: regulator -> pressure regulator

Added "pressure regulator"
P16 L358-360: and reducing the flow … -> and with an additional orifice to reduce the flow to about 55 sccm.

Now corrected by removing "and reducing the flow"

P16 L369: Use the term "Allan-Werle plot" in combination with Allan variance

We now used "Allan-Werle plot"

P16 L372: Define tau

We now defined Tau as "Tau is the abscissa of an Allan-Werle plot".

P16 L377: times scale - time scales

Time scales is now used

P17 L384: available Oxzilla fuel cell -> available fuel cell

The name "Oxzilla" is now removed

P17 L389: flow was adjusted to. It might be preferrable to use sccm as well.

Corrected as "Flow from each …"
P17 L391: flow out of the Oxzilla was … -> flow out of the fuel cell analyzer was

Corrected as "..flow out of the fuel cell analyzer…"

P17 L389++ : The use of different flow units is confusing. Please stick to sccm.

We now used sccm

P17 L397: It should be LabView instead of Lab VIEW. Could you specify the reason for using a labview program or is this not interesting ? Then drop this detail.

Corrected to "LabView"

P17 L398: I know "a priori", but what does "in priori" mean ? A priori wouldn't make no sense here, however.

We now used "First" instead of "In priori"

P19 L447+: "While similar correlation coefficients were observed for both analyzers, different slopes were calculated (Fig. A.1). This is due to the fact that the IRMS measures the O2 to N2 ratio ($\delta(O2/N2)$) in per meg, while the CRDS and the Paramagnetic analyzers provide non-calibrated O2 mixing ratios in units of ppm and per meg, respectively." Is this true ? I cannot come up with the observed slope values when I try to use the definition of per meg and ppm.

As we stated the difference is due to two reasons:

- Conversion factor between ppm and per meg
- Uncalibrated results from the Paramagnetic analyzer

So using only a conversion factor will not reproduce the observed slopes.

P18 L425: fractions -> mole fractions (change here and elsewhere)

Corrected as mole fractions

P18 L426: change and to or in "very low and very high O2".

Changed to "or"

P18 L426: standard 6 and 7 -> standards 6 and 7

Changed to standards 6 and 7

P18 L427: standard 7 was not measured on the IRMS ... change O2 mixing ratios to singular

Changed to "…the O2 mixing ratio is unknown"

P18 L429: between -> measured with

Corrected to …measured with…

P19 L432: lower in O2 -> lower

P20 L471: accounted -> accounted for

Corrected in line 501 now

P20 L474: in similar pattern -> following similar patterns

Corrected as suggested

P20 L477: The phrase repeats what has been said before. You could write. Interestingly the drift pattern could be modeled ...

Now modified as:

"Interestingly, the drift pattern can be modeled using a polynomial function which can…"

P20 L475: repeated use of drift. Drop the first in the sentence

Modified as "..Interestingly, this p[attern…."

P22 L506: Based on these plots -> Based on this plot

Now modified as "Based on this plot…"

P22 L506: I think that the essential features are reproduced but I don't think that there is very good agreement. Eg the NDIR shows an exponential response time when the CRDS is not.

We partly agree with the editor's reasoning, yet we have to bear in mind that we are talking water amounts in the very low ppm range. For these two analyzers that is the limit of their capabilities. Therefore, it was astonishing that the behavior is very similar with only very few exceptions. We would like to keep our statement since it also states the fact that we deal with low water contents.

P22 L509: measured in, not measured into

Deleted "to"

P22 L509 & P47 F15: The signals in panels a and b need to be explained. Where do the spikes come from and why does . Without such explanation it is impossible to follow your argument whether the agreement is good or not.

This is now included to the comments of Figure 15 below and in lines 540-549 in the manuscript.
P22 L512: in-built -> built-in

Corrected

P23 L545: "embedded within", check grammar

Now changed to "…installed inside.."

P24 L549: a low span -> low span ....

Corrected as low span

It is now modified as:

"Despite the strong variability, …"

P25 L555: It might be interesting to know which of the two devices was the first in the series.

Similar to the setup in Figure 11, the Paramagnetic system was first in the series.

Now it is modified as "…as it has not interference from possible CO2 absorption band overlap".

Deleted "if it appears"

Deleted "However"

Deleted "about" and the sentence is now read as:

"..we have observed a significant decrease in precision (about ten-fold) in…"

P29 L684: This conclusion is not identical to what is presented in the results section, where CO2 and CO are excluded.

This sentence is not contradicting the results section as it simply explains the components of breath air ($CO_2$, CO, $CH_4$ and $O_2$). For clarity we removed the line "… as breath air contains $CO_2$, $CH_4$ and CO in addition to oxygen." And the sentence is modified as:

"However, such measurements for a breath air showed a contrasting signal, possibly due to interference from other gases such as CH4."

Figure 8 is now updated as follows:

[Figure]

See above

P41 F9 : mention tau in legend (otherwise it is undefined in the figure caption)

Already explained in the main text as requested above

P46 F14: Please increase the size of the labels for CRDS and NDIR in the figure for readability
We have now modified this figure as follows

[Figure]

Figure 14. Parallel water vapor measurements for a dried ambient air by both the NDIR and CRDS

analyzers. Note that the water values from the NDIR analyzer are not calibrated.

P46 F14+ later on. SI units for mole fractions are mmol/mol, µmol/mol etc. The atmospheric community often uses permil, ppm etc for the same purpose. It would be better to be coherent and convert the NDIR water data to ppm, if this is the preferred (non-SI) unit of the authors. Using different units for the same quantity is confusing. The offset of the NDIR data is worrysome as it implies negativ values, which are physically impossible.

It is now converted into ppm.

The negative offsets in the NDIR are due to the uncalibrated water values as explained in the main text and Figure legend.
P46 F14: please shortly mention the reason why the NDIR analyzer always seems to relax into the stable O2 value after a humidity change whereas the CRDS does this only in a few cases.

This question is not clear plus the Figure does not display any O2 data.

P47 F15 : Please shortly explain origin of spikes in panels a and b.

These are standard gases measured in between samples and the spikes are due to valve switching.

P48 F16 : It would probably be interesting to have a histogram plot of the lower panel or a line fit (slope = 0 or different) to these data.

A horizontal line with slope of zero is now added.

[Figure]

P49 F17 : Figure is not in publishable quality. Axis labels on left hand side are cut. Also adapt label font to y- axis font (bold) and put units in parantheses.

This figure is now modified as shown below.

[Figure]

Figure 17. Diurnal variations of $CO_2$ (top) and $O_2$ (bottom) measurements from the 12 m (red) and the 212.5 m (black) height levels at Beromünster tower.

P50 F18 : Time scales of the two time series are different. Please align upper trace on the scale of the lower trace.
The figure is now adjusted as follows

[Figure]

P51 FA.1: Change "zooming only to standards 1-5" -> "selecting standards 1-5"

Changed to "selecting"

P51 FA.1: Convert all quantities into same units so that deviations from 1:1 correspondances can be spotted easily.

We kept these measurements in their respective units purposely, in case the reader is interested in converting the measured values in the multiple figures in this manuscript and to check any agreement/difference among these three analyzers. If we change the units, it will be difficult for the reader to easily check how we interconvert and compare the different results in the different plots.

P51 L910: Please change section title into something more specific. It is presently not very informative.

Removed the tittle "Additional plots"

[revised manuscript text omitted]

---

## Author Response (AR3)

**Second reply to the Editor´s comments**

We would like to thank the Editor for his constructive comments which we have addressed below. We keep the editor´s comments in red font and our replies in black font.

Dear authors,

Thank you for having addressed all different requests in your revised version of your manuscript.

Please find below a list of minor remarks. I also would like to ask you to include an equation that gives your measurand (O2/N2 or p(O2)/p) as a function of measured quantities and calibration gases. This is crucial to evaluate the influence of different parameters on the overall measurement uncertainty and it also avoids misconceptions due to a purely "textual" explication.

Minor comments

L 34 : exponent for year.

Now corrected as $yr^{-1}$

L 35 : Suggestion : Write "Atmospheric O2 is commonly expressed in units of per meg due to its small variability with respect to a large background, where ..."

This phrase is now re-written as:

"Atmospheric $O_2$ is commonly expressed in units of per meg due to its small variability with respect to a large background, where.."

L 42 - 43: I don't think that one can understand the phrase "Note that we convert per meg to parts per million equivalent by multiplying per meg by 0.209500 (the O2 mole fraction of atmospheric air)." Did you eventually mean "Note that we convert delta (O2/N2) values (in per meg) to mole fractions (in ppm) using the conversion factor 0.209500 (the O2 mole fraction of atmospheric air)." ? This is misleading. If I have delta = 0 per meg using the atmospheric ratio as a standard, I should have a mixing ratio of 0.209500 ppm. If I follow your description and multiply 0 by 0.209500, I get 0. It is preferable that you give an equation that unambiguously converts between the (dry) air mixing ratio and the delta values. The dry air mixing ratio clearly involves noble gases, CO2 and other minor components, whereas delta (O2/N2) does not ... I don't see how one can easily convert by a fixed number.

Regarding the conversion or proportionality between per meg and ppm levels, please have a look at either of the following Webpages.
Regarding zero per meg: This means that the sample corresponds to the standard ratio and since N2 is constant in the atmosphere (except changes due to seasonal temperature changes of the ocean), that the O2 of the sample corresponds to the O2 of the standard. Therefore, no deviation in O2 which corresponds indeed to zero ppm change.

See https://cdiac.ess-dive.lbl.gov/trends/oxygen/modern_records.html

The units of δ(O2/N2) are per meg; one per meg is one molecule of oxygen out of a million molecules of oxygen. Currently, this is roughly one molecule out of 4.77 million molecules of all gases in the atmosphere, not including water vapor molecules. Thus, 4.8 per meg is roughly one in a million molecules of dry air, or one part per million (ppm).

See http://scrippso2.ucsd.edu/faq
How does one relate ppm and per meg units?

These units refer to different types of quantities, so the question needs to be sharpened before it can be clearly answered. Suppose a tree consumes exactly one molecule of CO2 for each O2 molecule produced by photosynthesis. The changes in atmospheric O2 and CO2 near the tree will then be inversely proportional. What is the proportionality factor in per meg/ppm?

The answer is 1/.2095 = 4.8 per meg/ppm, where 0.2095 is the O2 mole fraction of air. This can be derived realizing that, because N2 is constant, the relative change in the O2/N2 ratio is the same as the relative change in O2 and calculating the relative change requires dividing by its abundance.

Therefore, the proportionality factor between per meg/ppm is
4.77 per meg = 1 ppm
per meg = 1 / 4.77 ppm = 0.209500. The latter number corresponds to the O2 mole fraction of air.

Actually, from eq. 1 you can easily see the proportionality if N2 is assumed to stay constant (fair assumption see above) and further write O2 of the sample as O2 of standard plus DO2. Resolving to DO2 yields the mentioned proportionality between per meg and ppm.
Also, please provide a reference for your value of 0.209500.

We have now provided the reference Machta and Hughes, 1970.
L 104: measurement wavelength -> wavelength measurement.

Now corrected accordingly

P 10 : As mentioned previously, the AFGL codes 161 and 162 are used exactly twice in the whole document (apart from the definition). Given that a large part of AMT readership has no spectroscopic background it is preferrable to use the more commonly known notation H2O and HDO (or the IUPAC definition) or the words "normal and deuterated water" at these instances. It has to be admitted that the AFGL notation is very handy for all people used to it.

We have now removed the AFGL codes and used $H_2O$ and HDO

L 151+: Phrase appears twice.

The repeated phrase is now removed

L 257 : I could not find the word "centration" in the dictionary. Do you mean "line centre position", perhaps ?

It is now corrected as "line center position"

L 260 : correct the exponent in wavenumber unit

It is now corrected as cm$^{-1}$

L 305 : Not clear what is meant by "the correlation may be due in part to the fitting procedure itself". You might need to explain a little bit more, even though a reference is given. I suspect (but I am not sure) that you speak about the choice of variables for the fitting function (rather than the procedure or the algorithm). For example, the width and the line area are necessarily correlated, whereas width and absorption depth are likely less correlated.

The editor's comments are correct. We meant to say that covariance of the width and area could be part of the explanation for the observed correlation. We now replaced "due in part to the fitting procedure itself" with "due in part to covariance of the fitted amplitude (proportional to line area) and line width".

L 267 : HITRAN gives a value gamma_H2O = 0.0514 cm^-1/atm, which seems to come from semi-emprical calculations : https://arxiv.org/abs/1906.01475 .
Please mention and check origin of that number.

We have now modified the paragraph from 265-272 as follows:

"From the linear fit one obtains a coefficient for collisional broadening of the Q13Q13 line by water vapor of γwater = 0.0442 cm-1/atm at 45 °C. Recently, parameters describing broadening of oxygen lines by water vapor, obtained by empirical modeling of selected experimental data, were added to the Hitran data base (Tan et al., In review). The new Hitran entries predict a value of 0.0486 cm-1/atm at 45 °C, which is in agreement with our measurement within the 5-10% uncertainty attributed by Hitran to the broadening parameter."

L 311 : "because the O2/N2 and water concentration", add "ratio" after "O2/N2" and "the" before "water concentration"

It is now corrected and read as:

".., because the O2/ N2 ratio and the water concentration affect the line width…"

L 329 : "normalizing absorption amplitudes to line widths does not provide any advantage". It is difficult to understand what is meant here. Could you please clarify further ? Do you mean normalising amplitudes by line widths ?

Yes, we mean normalizing amplitudes by line widths

We now replaced "to" with "by"

L 333 : Perhaps you should remove the word "Consequently" here. (As a reply to your comment on a similar and previous remark, please consider that the reader does not necessarily have Fig 1 at hand and in mind when comparing to the 16O2 absorption. Also, the noise level is not evident from that figure 1, as residuals usually reflect systematic effects such as bias in the profile, fringes, etc. some of which might scale with the size of the absorption line).

We have now removed the word "Consequently"

L 381 : write "tau is the averaging time" and use the greek symbol instead of the word "Tau"

The sentence in bracket is now modified as follows:

[revised manuscript text omitted]

---

## Author Response (AR4)

**Third reply to the Editor´s comments**

We would like to thank the Editor for his comments which we have addressed below. We keep the editor´s comments in red font and our replies in black font.

Dear authors,

Thank you for the clarifications and corrections. I would like to ask you some minor corrections.

As it stands, the phrase "Note that we convert per meg to parts per million equivalent by multiplying per meg by
0.209500 (the O2 mole fraction of atmospheric air) (Machta and Hughes, 1970)." is still incorrect or misleading at least. (Please also note that the title of the Machta and Hughes paper in the reference list contains a typographical error.)

The typo is now corrected to Atmospheric

First, parts "per million equivalent" seems to be science slang and the phrase should be rewritten as to explain the conversion between the measurement quantities "mole fraction" (commonly expressed in micromole/mole or ppm) and delta (commonly given in per meg units). Just talking about unit conversion could be misleading as the quantities that are converted into each other are completely different (we are not talking about conversion from inch to cm, for example).

We have now changed to sentence:
"Note that we convert per meg to parts per million equivalent by multiplying per meg by 0.209500 (the O2 mole fraction of atmospheric air)."

to

"Note that under the assumption that the atmospheric N2 content is constant (i.e. N2sample equals N2reference), we convert relative changes in oxygen given in per meg following equation 1 to oxygen changes in parts per million (equivalent to micromol/mol) by multiplying by the O2 mole fraction (O2reference) expressed as 209500 ppm. Hence 1 ppm corresponds roughly to 4.8 per meg, or 1 per meg to 1/4.8 (209500/10^6) ppm."

Second, the conversion factor is correct if we are concerned with relative changes, ie if delta changes by 1 per meg, then the O2 mole fraction changes by 0.2095 ppm. However, readers not familiar with O2 measurements or the delta notation will understand the sentence as 1 per meg corresponding to 0.2095 ppm and, correspondingly, that the mole fraction of O2 in air of ~ 0.2095 (or 209500 ppm) must correspond to 209500 per meg, which is obviously wrong. Since your article uses both quantities (mole fractions as well as delta values), the definition of quantities and units as well as the conversion formula should be given, preferably based on physical quantities with SI units. Note that AMT is a specialised journal dedicated to measurement techniques. Therefore, the correct definition of measurement quantities and their units is central and cannot be neglected.

See the comments above

Third, it is absolutely not trivial how the CRDS technology can measure delta (or O2/N2), because N2 is not measured (by that technology). The same holds for the mole fraction (O2/air). This is another reason why the measurand (absorption of O2) and its relation to the O2 mole fraction or the delta(O2/N2) should be clearly defined in the manuscript. Please explain the measurement process in more detail (what is measured, what is obtained from calibration and how).
Please understand that all this information must be given before the paper can be published.

The editor is correct that the instrument presented here measures the O2 mole fraction only. How the O2 concentration is obtained is clearly stated in the manuscript in chapter 2 followed by discussions on dependencies of the O2 mole fraction that have been thoroughly investigated as described
for instance in lines 311 to 319.

[revised manuscript text omitted]

---

## Author Response (AR5)

**Fourth reply to the Editor´s comments**

We would like to thank the Editor for his comments which we have addressed below. We keep the editor´s comments in red font and our replies in black font.

Dear authors,

Thank you for your corrections. Unfortunately, you haven't taken into account my remarks on the measurement principle and uncertainties.

Please take into account the following minor corrections.

1. Eq 5. Use a consistent notation for the mole fraction of gases, eg f_{O2}, f_{H2O} etc. ….

The functions for the retrieval of mole fractions are different for the different species since they depend on different spectral lines, but we understand the reviewers comment and rewrite eq. 5 as>

$O_{2,dry} = O_{2,raw}/(1-H_2O)$

where H2O is the measured water mole fraction

2. l. 321 It would be easier to understand if you use "multiplied" instead of "normalized".

We have now used the word "multiplied" as requested by the editor

3. The use of units still needs clarification. Since you use delta(O2/N2) in your paper, you should give an estimate of the error that is associated with the conversion from ppm to per meg. It is evident that the assumption of the N2 mole fraction being the same in the sample and reference is overly optimistic (l. 42 of your manuscript. As an aside note that it is not clear what you mean by N2 being constant. Partial pressure or mole fraction of N2 ?). There are natural variations of the partial pressures of CO2, Ar, H2O, etc. that require that if you maintain the same total pressure the mole fraction of N2 in sample and reference must differ at the ppm level, even if the amount of N2 in the atmosphere remains constant. It will be very helpful for the reader if you make the calculation in an appendix.
As I understand it, CRDS can "directly" measure the O2 mole fraction using a calibrated air-like gas mixture (which should not be too dry, otherwise the water correction scheme does not work properly). Conversion into delta(N2/O2) values, however, requires the knowledge of p(N2)/p(air), but from the measurement you know only p(O2)/p(air). Therefore you must make assumptions on Ar, CO2, H2O and all other substances that contribute to air at the ppm level. In this way you can calculate p(N2) = p(air)-p(O2) - p(all contributors at the ppm level).
What is the error of all these assumptions ? Please discuss/mention in the article.

The reviewer is correct that there is an influence of CO2, Ar, H2O or other atmosphere gas component changes on the mole fractions. We are well aware of this fact that has been dealt with in several publications beforehand. An easy way of understanding and estimating this influence can be obtained when noting all gas component of relevance as mole fractions and look at the influence of each when changing one component or several. We are happy to include the following in the Appendix if the editor finds this also helpful.

Addition to Appendix:

Influence of air composition on the measured $O_2$ and $dO_2/N_2$ following eq. 1.

The following equation describes exactly this issue:

[N2] + [O2] + [Ar] +[CO2] + [others] = 1 ;                              (A1)

where [] denote the mole fractions of the correspondent gas species (e.g. [N2] = 0.780840), all of which sums up to unity. Eq. (A1) can also be multiplied by 10^6 (one million) to express the mole fractions in ppm, e.g. [N2] = 0.780840 that corresponds to 780840 ppm.

If for instance [CO2] is changed to [CO2 + DCO2], DCO2 being a CO2 mole fraction change then eq. A1 changes to

[N2] + [O2] + [Ar] +[CO2+ DCO2] + [others] = 1 + [DCO2];            (A2)

This addition of DCO2 leads to a dilution of all other components correspondent to their mole fractions. Mathematically this is obtained

by dividing eq (A2) by the right-hand side term, e.g. 1 + [DCO2]. This leads to eq (A3)

([N2] + [O2] + [Ar] +[CO2+ DCO2] + [others])/(1 + [DCO2]) = 1;     (A3)

Example 1: CO2 change

From eq (A3) one can see that the ratios of any component with another component (e.g. O2/N2, Ar/N2 etc.) is not changing since the scaling factor (/(1 + [DCO2])) remains the same for the different components. In contrast the mole fractions indeed do change.

In the following we make an example for a DCO2 = 10 ppm

(10 ppm CO2 change) under the assumption that [N2] = 0.780840, [O2] = 0.209460, [Ar] = 0.009340, [CO2] = 0.000400, [others] = 0, The sum of them yields more than 1 and need to be normalized to get [N2] = 0.780809, [O2] = 0.209451, [Ar] = 0.009340, [CO2] = 0.000400, [others] = 0 by the corresponding division.

If now DCO2 = 10 ppm, then the sum of all components is 1.000010 and therefore the following mole fractions are obtained

[N2] = 0.780801, [O2] = 0.209449, [Ar] = 0.009340, [CO2] = 0.000410, [others] = 0

Using eq. 1 to convert it to O2/N2 leads to -10 permeg using [N2] = 0.780809 for both sample and reference as mentioned in the manuscript. DO2 corresponds to 2.1 ppm. This corresponds to 4.77 per meg change in O2/N2 per ppm O2 change.

Example 2: Ar change

In the following we make an example for a DAr = 100 ppm

(100 ppm Ar change) under the assumption that [N2] = 0.780809, [O2] = 0.209451, [Ar] = 0.009340, [CO2] = 0.000400, [others] = 0.

If now DAr = 100 ppm, then the sum of all components is 1.000100 and therefore the following mole fractions are obtained

[N2] = 0.780731, [O2] = 0.209430, [Ar] = 0.009439, [CO2] = 0.000400, [others] = 0

Using eq. 1 to convert it to O2/N2 leads to -100 permeg using [N2] = 0.780809 for both sample and reference. DO2 corresponds to 21 ppm. This corresponds again to 4.77 per meg in O2/N2 per ppm O2 change.

Generally, from these calculations one can see that 1 ppm change of any air component lead to a 4.77 per meg change in dO2/N2 after normalization to unity.

Therefore, it is indeed important to have additional information available for the air composition, in particular about CO2 and Ar mole fractions, otherwise O2 mole fractions or O2/N2 are misinterpreted.

Why do you compare to the standards using delta(O2/N2) instead of f(O2) even if that introduces an additional error ?

We made the comparison using delta(O2/N2) because this is the standard way in the community of expressing O2 variations.

---

## Author Response (AR6)

**Fifth reply to the Editor´s comments**

We would like to thank the Editor for his comments which we have addressed below. We keep the editor´s comments in red font and our replies in black font.

Dear authors,

Associate Editor Decision: Publish subject to minor revisions (review by editor) (24 Sep 2019) by Christof Janssen
Comments to the Author:
Dear authors,

Thank you for your corrections.

I think the quantitative difference between O2 mole fractions and the O2/N2 ratio is clear for the general reader and there is no need for the proposed appendix. What is less clear and what yet seems to be an inconsistency in the article is the following:

The abstract characterises the instrument performance in terms of O2 mole fraction and the uncertainty is expressed correspondingly. However, the article implies that it is rather the O2/N2 ratio that needs to be measured. This can be done easily when the composition of air is known. But if the composition of air is not known exactly, the relative uncertainty of O2/N2 is likely different from that of the O2 mole fraction alone and the uncertainty cannot be determined as easily. Please include a discussion/quantification of the additional factors/contribution to the uncertainty of O2/N2 such that the article can be published.

With best regards,

In order to clarify the above comment, we have now added the following paragraph at the end of section 3.2.1.
* * *

[revised manuscript text omitted]

---

## Author Response (AR7)

Reply to the two remaining comments of editor

Our reply is given in black font and the newly added paragraphs to the new version of the manuscript in italics, whereas the comments of the editor are marked in red.

Associate Editor Decision: Publish subject to minor revisions (review by editor) (11 Oct 2019) by Christof Janssen

Comments to the Author:

Dear Authors,

Thank you again for your mail pointing out that the review process is too slow. I hope to be as clear as possible so that any further unnecessary step can be avoided.

Generally speaking, I strongly approve your work and the article. It should certainly be published. However, the current version is inaccurate or erroneous in describing the link between mixing ratios and delta values. This needs to be clarified before publication. There are two points:

1. *A simple calculation using a dry three component model atmosphere of O2, N2 and others x = (Ar + ....) with relative abundances of 0.2094, 0.7809 and .0997 (TOHJIMA et al, Geophys Res Lett 30, 1653 (2003)) shows that the O2 mole fraction mf should change with a change dR in the R=O2/N2 ratio as d(mf) = (1+x)/(1 + R + x)^2 \* d(R) (assuming that all change in O2/N2 comes from OS and not from N2). The slope in the delta = d(R)/R over d(mf) diagram is thus given by R (1 + R + x)^2 (R(1+x))^-1. Plugging in numbers yields 6.04, which is close but different (why ?) from your value of 5.78 per meg per 1 ppm. The number of 4.8 per meg per 1 ppm on the first page of the manuscript is just wrong. Please correct the introduction and explain the origin of the factor 5.78 or 6.04 per meg per 1 ppm that readers can understand. You should provide details in an appendix. The section that you have added during the last revision refers to the wrong number of 4.8 per meg ppm^-1 and is superfluous at this section in the manuscript. Note that the examples that you have provided in the proposed appendix to manuscript version 7 did not include changes of O2 and contained an error. Your examples were concerned with the addition/dilution of the O2 mixing ratio by gases other than O2 and N2. It is correctly stated that this does not change the O2/N2 ratio, but changes O2 and N2 mole fractions. However in the ongoing calculation you then assume that the mole fraction of N2 remains constant while the mole fraction of O2 gets diluted. This certainly impacts the ratio, but the approach is self-contradictory and leads to the calculation of a meaningless number. Note that dilution cannot be used to determine a delta over d(mf) slope. When you calculate the change of the O2 mole fraction as a function of the dilution dx analogously to the above approach, you get d(mf) = 1/(1 + R + x)^2 \* d(x). However, since R does not depend on x, dR/dx=0. So the line traced by dilution by gases other than O2 or N2 should be horizontal (slope of 0 per meg per 1 ppm).*

2. *The paper lacks a dedicated statement on uncertainties related to the measurement of delta. All of the uncertainty discussion is concerned with mole fraction measurements, but since results need to be reported as delta values, the uncertainty of delta cannot be omitted. Using the above derived transfer coefficient from O2 mole fraction to delta, one can convert a mole fraction uncertainty of 5 ppm into*

*an equivalent value for delta. However, there are additional factors that impact delta (and don't have an effect on the O2 mole fraction), i.e. variability and lack of knowledge of the mole fraction of non-characterized atmospheric compounds. This is why the discussion needs to be extended. Please Mention these additional error sources and give a rough estimate of their contribution so that the reader can judge the degree of agreement with other measurement methods.*

Once these two issues are clarified, the paper can be published. Note that these issues have been raised in previous editor remarks, but have not been addressed. I am sorry that they have not been raised at or before the expert review stage, but only came up later in the review process.

Please take the time to carefully address the two above points so that any further revision becomes unnecessary. If the points are not addressed satisfactorily, I will send the revised version to one or more referees, whom I will ask to give a rapid and independent expert opinion on the issue.

With kind regards,

Christof Janssen

Dear Christof Janssen

We believe that the publication you are referring to in your comments is the publication of Tohjima et al. 2000 (JGR, Vol. 105, 2000) and not Tohjima et al. Geophys Res Lett 30, 1653 (2003), as the latter publication does not cover the points mentioned in your comments.

Equation 1 given in Tohjima et al. 2000 describes the total derivative of oxygen mole fraction. This is equivalent to what we have tried to describe in the proposed appendix to manuscript version 7, except that we obviously have missed to clearly state that we kept the nitrogen mole fraction constant since the Picarro analyzer is unable to measure $N_2$, Ar or $CO_2$. This of course, as you correctly mentioned, will lead to additional uncertainty due to the dilution effect of such changes. We have clarified these points in the following statements.

Following your view which is based on Thojima et al., 2000, it is correct that the per meg to ppm conversion has a slope of 6.04 if we consider only the changes in oxygen. It changes slightly to 6.11 per meg per ppm, when we talk about nitrogen changes only, and also as correctly stated in your statement above the slope is zero (horizontal line) when talking about any other changes of air components. The fact that our supplementary plot shows slopes of 5.78 (for the first five standards) or 6.08 (for all standards except ST-7) is due to a mixed influence from pure $O_2$, pure $N_2$ and other air component dilution effects. The lower slope of 5.78 documents particularly the influence of the $CO_2$ dilution effect.

Our view is consistent but follows another path, namely that the CRDS analyzer measures an $O_2$ concentration which requires to be converted to an $O_2/N_2$ ratio. Since no information about $N_2$ is available one assumes a constant value, i.e. $N_2$ of the standard. Therefore, eq. 1 in the manuscript reduces to $(O_{2,SA}/O_{2,ST}-1)*10^6$ or $\Delta O_2/O_{2,ST}*10^6$. The value obtained is an estimated $\delta O_2/N_{2,est}$ ratio, which indeed slightly different to the true $\delta O_2/N_{2,true}$. The effect of water dilution (amount of water vapor is measured by the CRDS analyzer) is taken into account as described in the manuscript. Yet any other dilution effect is not considered except if additional information is available, e.g. $CO_2$ concentration measurements. Indeed this dilution effects can be significant and are displayed in the following table.

| Change in ppm | | $\Delta O_2$ apparent (change + dilution effect) | $\delta O_2/N_{2,true}$ (per meg) ($\delta O_2/N_{2,true}/\Delta O2$ in per meg/ppm) | $\delta O2/N_{2,est}$ (per meg) ($\delta O2/N_{2,est}/\Delta O2$ in per meg/ppm) | Difference in $\delta O2/N2$ (
[revised manuscript text omitted]

Furthermore, we changed the text in the introduction section referring to equation 1 which was written as:

" Note that under the assumption the atmospheric N2 content is constant (i.e. N2sample equals N2reference), we convert relative changes in oxygen given in per meg following equation 1 to oxygen changes in parts per million (equivalent to micromol/mol) by multiplying by the O2 mole fraction (O2reference) expressed as 209500 ppm. (the O2 mole fraction of atmospheric air) (Machta and Hughes, 1970). Hence 1 ppm corresponds approximately to 4.8 per meg, or 1 per meg to 1/4.8 (209500/106) ppm."

*New text in manuscript after eq. 1*

*Equation 1 is used to convert oxygen mole fraction changes expressed in ppm (as measured by several techniques such as paramagnetic cell, UV-cell as well as the by the CRDS analyzer presented here) into changes in $\delta O_2/N_2$. This is associated with the influence of dilution effects on the mole fractions but not necessarily on the ratios. These conversion difficulties and their expressions in uncertainties are discussed in the Appendix.*

---

## Author Response (AR8)

**Revision:**

**We thank for the careful reading of the revision and hope that we meet with our revision of the Appendix the expectations.**

**Associate Editor Decision: Publish subject to minor revisions (review by editor)** (11 Nov 2019) by Christof Janssen
Comments to the Author:
Dear Authors,

Thank you very much for your response. I appreciate the level of detail that you dedicate to the discussion of the conversion from O2 mole fraction to delta(O2/N2). I fully agree with your suggestion of modifying the introduction and to move the detailed discussion on the relation between O2 mole fraction and delta(O2/N2) to the Appendix. The first two paragraphs are very comprehensible. You could state before "Following equation 1 in Thojima ..." that no unique conversion factor exists if N2 or the other air components are not known.

Unfortunately, paragraphs 3 and 4 of the analysis and table A1 in the Appendix still lead to confusion and the two paragraphs need to be corrected/rewritten. I cite:

"Note that under the assumption the atmospheric N2 content is constant (i.e. N2sample equals N2reference), we convert relative changes in oxygen given in per meg following equation 1 to oxygen changes in parts per million (equivalent to micromol/mol) by multiplying by the O2 mole fraction (O2reference) expressed as 209500 ppm (the O2 mole fraction of atmospheric air) (Machta and Hughes, 1970). Hence 1 ppm corresponds approximately to 4.8 per meg, or 1 per meg to 1/4.8 (209500/106) ppm.
This is used in our approach since the Picarro ... in the following Table A1"

The above text seems to indicate that in your calculation you consider changes to O2 alone (thus keeping N2, CO2, Ar, ... const). If done correctly, this corresponds to case 3 in your table A1 ($\Delta$O2 only), which indicates that the conversion factor is 6.04 per meg /ppm, but you derive 4.8 per meg /ppm from your calculation ! It is evident that there is an error or an undocumented approximation somewhere in your calculation. This contradiction needs to be resolved.

I might be wrong, but from your arguing, you take the value of 4.77 per meg / ppm as an ad hoc conversion factor somewhere in the range of values that might be observed (between 0 and 6.11 per meg / ppm). If this is true, you should present it in exactly this way. If this is not the case, please give another motivation that does not contradict the results in your Table A1.

Finally you mention the N2 mole fractions of your standards but you don't give them in Table 1. However, this seems to be very important information. With N2, O2, CO2 given, using reasonable Ar values and a measured data for H2O, much of the uncertainty in the conversion from mole fraction to delta should be very small. Please add these data if they are available.

We, have added a new table A1 that lists all the information for the scenarios displayed in table A2 for the conversion of mole fractions in delta O2/N2.

Minor corrections/suggestions

Following equation 1 in Thojima et al., 2000, the per meg to ppm conversion has a slope of 6.04 if only a change in oxygen is applied as seen in table A1.
->
Following equation 1 in Thojima et al., 2000, the per meg to ppm conversion has a slope of 6.04 if a change only in oxygen is applied as seen in table A1.

Corrected as suggested above

The fact that our supplementary plot shows slopes of 5.78 (for the first five standards) or 6.08 (for all standards except ST-7) is due to a mixed influence dilution effects.
->
The fact that our supplementary plot shows slopes of 5.78 (for the first five standards) or 6.08 per meg per ppm (for all standards except ST-7) is due to a mixed influence of dilution effects.

Corrected as suggested above

The lower slope of 5.78 documents particularly the influence of the $CO_2$ dilution effect.
->
The lower slope of 5.78 per meg per ppm documents particularly the influence of the $CO_2$ dilution effect.

Corrected as suggested above

For example a 10 ppm increase in $CO_2$ lead to an incorrect value of -10 per meg in $\delta O_2/N_2$,est compared to the true $\delta O_2/N_2$,true value.
->
For example a 10 ppm increase in $CO_2$ leads to a bias of -10 per meg in $\delta O_2/N_2$,est compared to the true $\delta O_2/N_2$,true value.

Corrected as suggested above

As you can see from table A1 an addition of N2 of 10 ppm leads to a reduced and opposite effect for the difference in $\delta O_2/N_2$ (true – est) because the dilution effect on O2 cannot compensate the change from the increase in nitrogen, therefore it scales with -10 ppm x (oxygen mole fraction/nitrogen mole fraction).
->
As can be seen from table A1, addition of 10 ppm of N2 leads to a reduced and opposite effect for the difference in $\delta O_2/N_2$ (true – est) because the dilution effect on O2 cannot compensate the change from the increase in nitrogen.

Corrected as suggested above

In our case this is given – but can certainly be improved – since we are comparing air composition to air standard compositions.
->
In our case this is given – but can certainly be improved – since we are comparing natural air to air standards.

Corrected as suggested above

Yet, determinations of the standards that has been used in this study have a range in N2 concentrations of -110 to +110 ppm for the ST-1 to ST-5, whereas ST-6 (+700 ppm) and ST-8 (-6200 ppm) are significantly off compared to our primary standard used for mass spectrometric determination.
->
Yet, determinations of the standards that have been used in this study have a range in N2 concentrations of -110 to +110 ppm for the ST-1 to ST-5, whereas ST-6 (+700 ppm) and ST-8 (-

6200 ppm) are significantly off compared to our primary standard used for mass spectrometric determination.

Corrected as suggested above

Table A1: ppm to per meg conversion calculations for air-like compositions
->
Table A1: mole fraction to delta conversion for air-like gas compositions according to different scenarios

Corrected as suggested above

$\delta O2/N2$,true: for instance measured by mass spectrometry; $\delta O2/N2$,est: for instance measured by 988 Picarro G-2207.
->
$\delta O2/N2$,true: calculated (eg Thojima et al., 2000); $\delta O2/N2$,est: use fixed conversion factor of 4.77 per meg / ppm

Corrected as suggested above

$\Delta O2$ in line 4 of your Table A1 should be $\Delta N2$

Corrected as suggested above

New Appendix:

*Appendix: Uncertainty consideration during mole fraction (ppm) to delta (per meg) conversion for air-like gas compositions according to different scenarios*

*Generally, the Delta notation, as given in equation 1 of this publication (main text), is used in order to circumvent the influences of dilution by other gas components when determining oxygen mole fractions (see table A1). Yet, several instruments are measuring the oxygen mole fraction such as the paramagnetic cell, the UV-cell as well as the instrument by Picarro presented here. Therefore, a thorough consideration of the conversion from ppm (mole fraction) to per meg (Delta O2/N2 notation) is necessary which we do in this appendix.*

*Following equation 1 in Thojima et al., 2000, the per meg to ppm conversion has a slope of 6.04 if a change only in oxygen is applied as seen in table A2. It changes slightly to 6.11 per meg per ppm, when we talk about nitrogen changes only, or to a slope zero (horizontal line) when talking about any other changes of air components excluding oxygen and nitrogen. The fact that our supplementary plot shows slopes of 5.78 (for the first five standards) or 6.08 per meg per ppm (for all standards except ST-7) is due to a mixed influence dilution effects. The lower slope of 5.78 per meg per ppm documents particularly the influence of the CO2 dilution effect.*

$$\delta \left(\frac{O_2}{N_2}\right)(per\ meg) = \left(\frac{(\frac{O_2}{N_2})_{sample}}{(\frac{O_2}{N_2})_{reference}} - 1\right) \cdot 10^6 \quad (1)$$

Note that under the assumption the atmospheric N2 content is constant (i.e. N2sample equals N2reference), we convert relative changes in oxygen given in per meg following equation 1 to oxygen changes in parts per million (equivalent to micromol/mol) by multiplying by the O2 mole fraction (O2reference) expressed as 209500 ppm. (the O2 mole fraction of atmospheric air) (Machta and Hughes, 1970). Hence 1 ppm corresponds approximately to 4.8 per meg, or 1 per meg to 1/4.8 $(209500/10^6)$ ppm.

This is used in our approach since the Picarro instrument gives us an O2 mole fraction which requires to be converted to an O2/N2 ratio. Since no information about N2 is available one assumes a constant value, i.e. N2 of the standard. Therefore, eq. 1 in the manuscript reduces to $(O_{2,SA}/O_{2,ST}-1)*10^6$ or $\Delta O_2/O_{2,ST}*10^6$. The value obtained is an estimated $dO2_{,norm}/N2_{,base}$ ratio, which indeed does not need to correspond to the true $dO2_{,norm}/N2_{,norm}$. The effect of water dilution (amount of water vapor is measured by the Picarro 2207 instrument) is taken into account as described in the manuscript. Yet any other dilution effect is not considered except if additional information is available, e.g. CO2 mole fraction measurements. Indeed this dilution effects can be significant and are displayed in the following table A2.

Table A1: Different dilution scenarios on their effects on air-like compositions

| Mole fraction | base | original | normalized | original | normalized | original | normalized | original | normalized |
|---|---|---|---|---|---|---|---|---|---|
| N2, ppm | **780809** | 780809 | 780801.192 | 780809 | 780801.192 | 780809 | 780801.192 | 780819 | 780811.192 |
| O2, ppm | 209451 | 209451 | 209448.906 | 209451 | 209448.906 | 209461 | 209458.905 | 209451 | 209448.906 |
| Ar, ppm | 9340 | 9340 | 9339.9066 | 9350 | 9349.9065 | 9340 | 9339.9066 | 9340 | 9339.9066 |
| CO2, ppm | 400 | 410 | 409.9959 | 400 | 399.996 | 400 | 399.996 | 400 | 399.996 |
| Total | 1000000 | 1000010 | 1000000 | 1000010 | 1000000 | 1000010 | 1000000 | 1000010 | 1000000 |
| | base | original | normalized | original | normalized | original | normalized | original | normalized |
| | | | | | | | | | |
| Change | | original | apparent | original | apparent | original | apparent | original | apparent |
| DCO2, ppm | | 10 | 9.9959 | 0 | -0.0040 | 0 | -0.0040 | 0 | -0.0040 |
| DAr, ppm | | 0 | -0.0934 | 10 | 9.9065 | 0 | -0.0934 | 0 | -0.0934 |
| DO2, ppm | | 0 | -2.0945 | 0 | -2.0945 | 10 | 7.9054 | 0 | -2.0945 |
| DN2, ppm | | 0 | -7.8080 | 0 | -7.8080 | 0 | -7.8080 | 10 | 2.1919 |

This will lead to the following changes for oxygen, the true and estimated (when N2 is not measured) dO2/N2 ratios and the difference in these dO2/N2 values.

Table A2: Mole fraction to delta conversion for air-like gas compositions according to different scenarios

| Change in ppm | | DO2 apparent (change + dilution effect) | dO2,norm/N2,norm (per meg) (dO2,norm/N2,norm/DO2 in per meg/ppm) (true) | dO2,norm/N2,base (per meg) (dO2,norm/N2,base/DO2 in per meg/ppm) (estimated) | Difference in dO2/N2 (true – estimated) in per meg |
|---|---|---|---|---|---|
| DCO2 only | 10 | -2.0945 | 0 (0) | -10 (4.77) | 10 |
| DAr only | 10 | -2.0945 | 0 (0) | -10 (4.77) | 10 |
| DO2 only | 10 | 7.9054 | 47.74 (6.04) | 37.74 (4.77) | 10 |
| DN2 only | 10 | -2.0946 | -12.81 (6.11) | -10 (4.77) | -2.81 |

$dO_{2,norm}/N_{2,norm}$: calculated (eg Thojima et al., 2000) or measured for instance by mass spectrometry; $dO_{2,norm}/N_{2,base}$: use fixed conversion factor of 4.77 per meg / ppm based on measured O2 mole fraction (ppm) for instance by Picarro 2207.

Incorrectly assumed N2, Ar, CO2 or any additional gas component lead to changes in the estimated $dO2/N2_{,est}$ values as stated in the table A2. For example a 10 ppm increase in CO2 leads to an incorrect value of -10 per meg in $dO_{2,norm}/N_{2,base}$ compared to the true $dO_{2,norm}/N_{2,norm}$ value. This is

*simply the dilution effect that the increased CO2 mole fraction has on the correspondingly measured O2 mole fraction (Table A1) (dilution in oxygen corresponds to the percentage-wise assignment of the excess CO2 in ppm to oxygen, i.e. -10 ppm x oxygen mole fraction = 2.0945, if O2 mole fraction corresponds to 0.20945). As you can see from table A1, addition of N2 of 10 ppm leads to a reduced and opposite effect for the difference in dO2/N2 (true – estimated) because the dilution effect on O2 is not able to compensate the change from the increase in nitrogen, therefore it scales with -10 ppm x (oxygen mole fraction/nitrogen mole fraction). This also tells us that the difference in the Delta values (true – est) scales with the O2/N2 ratio present in the sample. Therefore, best results are obtained when the calibration gases for which the gas composition is known equals closely the sample gas composition. In our case this is given – but can certainly be improved – since we are comparing natural air to air standard compositions. Yet, determinations of the standards that have been used in this study have a mole fraction range in N2 of -110 to +110 ppm for the ST-1 to ST-5, whereas ST-6 (+700 ppm) and ST-8 (-6200 ppm) are significantly off compared to our primary standard used for mass spectrometric determination. Therefore, special attention is required for the precise determination of standard gas composition and the control of the air sample composition by means of flask measurements in order to detect potential fractionation effects during air intake.*

---

## Author Response (AR9)

We could like to thank the Editor for his constructive and useful comments. We have now incorporated the comments below into the new manuscript and provided the replies point by point below.

Associate Editor Decision: Publish subject to technical corrections (26 Nov 2019) by Christof Janssen
Comments to the Author:
Dear Authors,

Thank you for the revised version of your manuscript which clarifies the raised issues. I therefore congratulate you to your very interesting paper that can be published subject to technical corrections.

To eliminate eventual ambiguities, I propose to delete
- "following equation 1" in the phrase "Note that under the assumption the atmospheric $N_2$ content ..."

This phrase is deleted as proposed above

- the three phrases from "Since no information about $N_2$ is available ...." to "correspond to the true $dO_2norm/N_2norm$." (because all these quantities have not been defined yet and potentially create more confusion than they explain)

These phrases are now deleted
- "the following" in the phrase beginning with "Indeed this dilution effects"

This phrase is now deleted

You should also replace "this" by "these" in the same phrase and "addition of $N_2$ of 10 ppm" by "addition of 10 ppm $N_2$" at a later instance.

"This" is now replaced with "these" and modified as proposed above